# Bioengineered phytomolecules-capped silver nanoparticles using *Carissa carandas* leaf extract to embed on to urinary catheter to combat UTI pathogens

Haajira Beevi Habeeb Rahuman[1,☯], Ranjithkumar Dhandapani[1,☯], Velmurugan Palanivel[2,*], Sathiamoorthi Thangavelu[1], Ragul Paramasivam[3], Saravanan Muthupandian[4,5,*]

**1** Department of Microbiology, Science Campus, Alagappa University, Karaikudi, Tamilnadu, India, **2** Centre for for Material Engineering and Regenerative Medicine Bharath Institute of Higher Education, Chennai, India, **3** Chimertech Innovations LLP, Tamilnadu Veterinary and Animal Science University, Chennai, India, **4** Division of Biomedical sciences, College of Health Sciences, School of Medicine, Mekelle, Ethiopia, **5** AMR and Nanomedicine Laboratory, Department of Pharmacology, Saveetha Dental College, Saveetha Institute of Medical and Technical Sciences (SIMATS), Chennai, India

☯ These authors contributed equally to this work.
* bioinfosaran@gmail.com (SM); palanivelmurgan2008@gmail.com (VP)

**Data Availability Statement:** All relevant data are within the paper and its Supporting information files.

## Abstract

Rising incidents of urinary tract infections (UTIs) among catheterized patients is a noteworthy problem in clinic due to their colonization of uropathogens on abiotic surfaces. Herein, we have examined the surface modification of urinary catheter by embedding with eco-friendly synthesized phytomolecules-capped silver nanoparticles (AgNPs) to prevent the invasion and colonization of uropathogens. The preliminary confirmation of AgNPs production in the reaction mixture was witnessed by the colour change and surface resonance plasmon (SRP) band at 410nm by UV–visible spectroscopy. The morphology, size, crystalline nature, and elemental composition of attained AgNPs were further confirmed by the transmission electron microscopy (TEM), selected area electron diffraction (SAED), X-ray diffraction (XRD) technique, Scanning electron microscopy (SEM) and energy dispersive spectroscopy (EDS). The functional groups of AgNPs with stabilization/capped phytochemicals were detected by Fourier-transform infrared spectroscopy (FTIR). Further, antibiofilm activity of synthesized AgNPs against biofilm producers such as *Staphylococcus aureus*, *Escherichia coli* and *Pseudomonas aeruginosa* were determined by viability assays and micrographically. AgNPs coated and coating-free catheters performed to treat with bacterial pathogen to analyze the mat formation and disruption of biofilm formation. Synergistic effect of AgNPs with antibiotic reveals that it can enhance the activity of antibiotics, AgNPs coated catheter revealed that, it has potential antimicrobial activity and antibiofilm activity. In summary, *C. carandas* leaf extract mediated synthesized AgNPs will open a new avenue and a promising template to embed on urinary catheter to control clinical pathogens.

**Funding:** The authors received no specific funding for this work.

**Competing interests:** The authors have declared that no competing interests exist.

## Introduction

UTI is broadly defined as a symptomatic or asymptomatic infection in both upper and lower urinary system which involves initial adhesion and colonization on the surface of the medical devices (catheter). The bacteria implicated in UTIs are *Staphylococcus sp*., *Streptococcus sp*., *Klebsiella sp*., *Enterococcus sp*., *Proteus sp*., *Pseudomonas sp*., and *Escherichia coli* owing to its biofilm assembly capacity [1–3]. Among most of the UTI cases, 80% are allied with ingrained urinary catheters [4] and associated UTIs are foremost common infection throughout the world [5]. The colonization of microbial community on medical devices forms a polymicrobial aggregates called "biofilm". Self-generated extracellular polymeric matter adheres the surface of the hospital acquired devices give rise to implant failure. It has been accounted that to control biofilm forming bacteria needs 1500 times higher concentration of antibiotics when compared to planktonic bacteria [6]. The existence of urine in urinary catheters makes an appropriate habitation for urease-positive microbes. The pH of the urine increases due to the presence of ammonia which makes the deposition of calcium and magnesium phosphate on catheter can ultimately leads to thorough constriction of the biofilm on catheter over coating or crystalline biofilms [7]. The UTI bacteria cause serious concerns due to spreading to kidney and cause acute or chronic pyelonephritis [8]. Increased antibiotic resistance of biofilm was formed by extracellular polymeric substances (EPS) matrix, found in the biofilm communities which makes the treatment ineffective [9]. A review by [8] Singha et al., 2017 described the several attempts have been made to impregnating antimicrobial coating on catheter with anti-biotics, antimicrobial agents (both biocidal and antifouling), antimicrobial peptides, bacterio-phages, enzymes, nitric oxide, polyzwitterions, polymeric coating modifications, liposomes. These coating have shown good antimicrobial activity *in vitro*, however a few drawbacks are shortlisted including resistance development. Silver nanoparticles produced from the phyto-chemicals of *C. carandas* leaf extract have been studied as a major and promising antibacterial alternative and also inhibit the biofilm formation in UTI pathogens. It was used as an antimi-crobial nanomaterial for coat urinary catheter in order to prevent catheter associated UTI infection.

Among the various inorganic metal nanoparticles, silver nanoparticles (AgNPs) have gained its attention for various reasons such as low toxicity, environment friendly and also known for its antibacterial activity against the bacteria exhibiting resistance to antibiotics [10]. Silver exhibits excellent antimicrobial activity and the production of nanomaterial through physical and chemical approaches will have an adverse effect in environment due to the adsorption of toxic substance as a reducing agent [11]. The system of phytochemical mediated synthesis of nanomaterial is a promising eco-friendly, non-toxic, cheap substrate, easily available, convenient and quickly processable to fabricate antimicrobial nanomaterial [11,12]. *C. carandas* belongs to the species of flowering shrub in dogbane family, Apocyana-ceae. *Carissa carandas* spread widely throughout the tropical and subtropical region of India. The plant possessing phytochemical constituents has high medicinal values [13]. In tradi-tional medicine, *Carissa carandas* leaf, bark, fruit, root have been used to treat several human ailments such as hepatomegaly, indigestion, amenorrhea, oedema, colic, piles, antipyretic, fever, liver dysfunction, stomach pain, skin infections, intestinal worms, antimicrobial, anti-fungal [14–16]. The leaf of *C. carandas* has anticancer, antimicrobial, antioxidant property and non-mutagenic property [17]. The leaf decoction is used to treat against sporadic fever, remedy for diarrhea, earache, syphilitic pain, oral inflammation and snake bite poisoning [18]. Since this plant has many medicinal values and very less literature availability for *C. car-andas* leaf extract.

In this research, the leaf extract of *C. carandas* was used to reduce the precursor solution of silver nitrate to AgNPs and this production was optimized by modifying parameters of synthesis such as pH, *C. carandas* leaf extract, metal ion concentration, and production time. Characterization of synthesized AgNPs was done by UV Vis spectrophotometry, TEM, XRD, EDS, FTIR and SAED pattern. The synthesized AgNPs was investigated for antimicrobial activity and embedded on catheter to investigate the property as antimicrobial nanomaterial to inhibit catheter associate UTI infection.

## Materials and method

### Chemicals and biological materials

Fresh leaves of *C. carandas* were collected from Periyakulam, Theni District, Tamilnadu, India (10.1239° N, 77.5475° E) and washed thoroughly to remove the dust. Silver nitrate ($AgNO_3$), Muller Hinton Agar (MHA), Lysogenic broth (LB), trypticase soya broth (TS) was acquired from Hi-media and used to assess antibacterial, antibiofilm assays. Bacterial pathogens such as *Escherichia coli* AMB4 (MK788230), *Pseudomonas aeruginosa* AMB5 (clinical sample), *Staphylococcus aureus* AMB6 (Clinical sample) was maintained by Department of Microbiology, Alagappa University, Science campus, Karaikudi, India.

### Extract preparation

Cleaned *C. carandas* leaves were subjected to air dry and quantified the weight of 100 grams. Dried leaves were soaked in 300 mL of Millipore water and allowed to boil for 1 h at 80°C to avail decoction of leaf extract which was percolated through Whatmann no.1 filter paper and stored at 4 °C for future use.

### Synthesis and optimization of AgNPs production

The AgNPs synthesis was carried out by adding 1mL of filtered *C. carandas* leaf extracts and 9mL of 1.25mM aqueous silver nitrate solution ($AgNO_3$) in the ratio of 1:9 was incubated at ambient temperature under dark condition. Initial AgNPs production was confirmed by visual color change from light yellow to dark brown color and scanning the absorbance along the UV-Vis range (200–600 nm) of the electromagnetic spectra using an UV-Visible Spectrophotometer (Shimadzu UV 1800, Japan). To achieve large scale production of AgNPs, optimization procedure was followed by modifying the parameters like pH, substrate (extract), metal ion concentration and production time. Briefly, pH of the solution was optimized by modifying the solution to various pH 2, 3, 4, 5, 6, 7, 8, 9, 10 with 1mL substrate (extract) concentration and 0.1mM metal ion concentration, left overnight under dark condition. Substrate concentration was optimized by modifying the solution to various concentration like 0.1, 0.5, 0.75, 1, 1.25, 1.5, 1.75 mL with the optimized pH as a standard and 0.1mM metal ion concentration, left overnight under dark condition. $Ag^+$ ion concentration was optimized by modifying the solution to various metal ion concentration such as 0.25, 0.5, 0.75, 1, 1.25, 1.5, 1.75, 2, 2.25, 2.5mM with the optimized pH and optimized substrate concentration, left overnight under dark condition. Then finally production time was optimized by measuring the absorbance at various time intervals such as 0, 5, 10, 15, 20, 25, 30 mins with the optimized pH, substrate and metal ion concentration using UV-Visible Spectrophotometer. With the optimized parameters the optimum production was set for the large-scale production. The heterogeneous mixture was centrifuged at 12000 rpm for 20 min followed by collection of pellets; washed with methanol: water ratio at 6:4 and lyophilized to obtain nanoparticles powder.

## Characterization of nanoparticles

XRD (X-ray diffraction) analysis of silver nanoparticles was recorded by P analytical X' Pert PRO powder which was operated at a voltage of 40kV with the current of 30 mA using Cu-Kα radiation of wavelength 1.5406 Å in the 2θ range of 20˚- 80˚ to obtain the crystalline structure of the AgNPs. Involvement of functional group in synthesis of nanoparticles and capping material was monitored by FTIR (Fourier Transform Infrared spectrophotometer) and performed to analyze the presence of functional groups of AgNPs and capping phytochemicals using attenuated total reflectance (ATR) mode (Nicolet iS5, Thermo Fisher Scientific Inc., Marietta, GA, USA). EDX (Energy dispersive X-ray) analysis was performed to determinate the elemental composition (Tescan VEGA 3SBH with Brukar easy). HR-TEM (High resolution Transmission Electron microscope) (JEOL-2100+, Japan) and SAED (selected area Electron Diffraction) pattern were analyzed to examine the size, crystalline structure and surface morphology of AgNPs.

## Antibacterial activity

Each test bacterial strain of 0.5 McFarland standards [19] was swabbed on MHA plates using a sterile swab and a well of 8mm width was formed using a sterile well borer under aseptic condition. Different concentrations of AgNPs 25, 50, 75, 100, 125μg/mL (1mg/mL stock solution was prepared for synthesized AgNPs, from the stock solution 25 μL was dissolved in 975 μL of DMSO to make 25μg/mL concentration and further concentrations were prepared accordingly) were loaded in the MHA plates along with the DMSO as solvent control and incubated at 37ºC for 24 h. After incubation, zone of inhibition (ZoI) was measured to the nearest millimeter from end of the well to end of the zone.

Comparison was made with AgNPs (125 μg/mL), crude leaf extract (50μg/mL), 1.25mM AgNO$_3$ solution, 99.8% of DMSO as a solvent negative control and ciprofloxacin (50μg/mL) as positive control for assessment were loaded consequently in the agar wells made in MHA plate and incubated at 37ºC for 24 h. After incubation, zone of inhibition (ZoI) was measured to the nearest millimeter from end of the well to end of the zone.

## Minimum Inhibitory (MIC) and Minimum Bactericidal Concentration (MBC)

The MIC and MBC was performed to evaluate the efficiency of obtained AgNPs to inhibit bacterial pathogens and protocol was followed according to the guidance of CLSI. MIC was performed by 96 microtiter well plate by broth micro dilution method. $10^6$CFU/mL concentration of bacterial inoculum (10μL) was inoculated with different concentrations of AgNPs (20, 40, 60, 80, 100, 120, 140, and 160μg/mL) and incubated at 37 ºC for 24 h. After incubation well plates were recorded by ELISA reader at 590nm to assess it optical density value. MIC was analyzed to determine the efficacy of appropriate concentration of AgNPs required inhibiting the bacterial growth. The inhibition rate can be estimated as follows

$$\% \text{ Inhibition rate} = 100 \times \frac{(\text{OD}_{\text{untreated}} - \text{OD}_{\text{well}})}{(\text{OD}_{\text{untreated}} - \text{OD}_{\text{blank}})} \tag{1}$$

Where OD$_{\text{untreated}}$ = optical density of bacterial cell without AgNPs, OD$_{\text{well}}$ = optical density of bacterial cell with AgNPs, OD$_{\text{blank}}$ = sterile culture medium.

The MIC endpoint is the lowest concentration of silver nanoparticles where no visible growth is seen in the well. The visual turbidity was noted, both before and after incubation of well plate to confirm MIC value [20].

After incubation the titer plates were agitated gently for 10 min and the broth in the well were plated on MHA plate and incubated during 24h, the CFU was counted and bacterial viability was calculated in order to calculate the MBC. MBC cutoff occurs when 99.9% of the microbial population is destroyed at the lowest concentration of AgNPs [20].

## Synergistic effect of silver nanoparticles with commercial antibiotics

Synergistic effect of silver nanoparticles with commercial antibiotics for uropathogens was done by disk diffusion method. Commercial antibiotic discs were impregnated with synthesized AgNPs (Ciprofloxacin -50mcg, Trimethoprim– 30 mcg, Gentamycin– 30 mcg) in the concentration of 20μg/mL and allowed to air dry. Then MHA plates were prepared and inoculated with overnight bacterial culture in the turbidity of 0.5% of McFarland standard. Commercial antibiotic disc impregnated with AgNPs was placed on the MHA plates and control plates were swabbed with test culture and placed with commercial discs aseptically. These plates were incubated at 37 ºC for 24 h and the zone of inhibition was measured [21].

## Qualitative assay for biofilm formation

Qualitative assessment of the pathogen's biofilm potential was performed by test tube method according to [7] Doll et al., 2016. Briefly, trypticase soy broth was inoculated with loop full of mid-log phase pathogen and incubated at 37˚C for 24 h. Uninoculated broth was considered as a control. The broth was removed 24 hours of incubation and tubes were cleaned with sterile Phosphate buffered saline PBS with the pH of 7.4. The tubes were dried and stained for 10 minutes with 0.1 percent crystal violet. Extra dye was removed with sterile distilled water and stained film formed at the tube's base, indicating the development of biofilm [22].

## Quantitative assay for biofilm formation

Development of static biofilm formation was confirmed by quantitative assay by microtiter plate method. Mid-log phase culture was diluted ten times using a sterile media. The culture was transferred to microtiter plate. The plates were incubated at 37 ºC for 16h. After incubation, planktonic cells were removed using PBS (pH 7.2) and dried, subsequently the plates were stained with 125μL of 0.1% CV solution. Dye in the well surface was solubilized using 200μL of 30% glacial acetic acid, the content of each well was mixed and transferred to sterile well plate and this setup was read at 590nm. The test organisms were classified as weakly, moderately adherent, non-adherent and strongly adherent bacteria based on the criteria ($OD < OD_C$ = Non adherent, $OD_c < OD < 2 \times OD_C$ = weakly adherent, $2 \times OD_c < OD < 4 \times OD_c$ = moderately adherent, $4 \times OD_c < OD$ = strongly adherent where $OD_c$ = average OD of negative control [23].

## Coating of urinary catheter with AgNPs

Urinary Catheter was segmented to 1×1cm. Catheter pieces were entirely dipped in synthesized AgNPs suspension with different concentration of AgNPs coated catheter such as 20μg/mL, 40μg/mL, 80μg/mL, 120μg/mL, 160μg/mL for 24 h. Excess of suspension was removed by blotting and dried at 50ºC [24].

## Biofilm inhibition in AgNPs coated catheter

Conical flask containing 25mL of sterile trypticase soy broth inoculated with 100 μL of mid-log phase pathogenic culture. Two sterile catheters were introduced into the medium using sterile forceps. The different concentration of synthesized AgNPs coated catheter (20μg/mL,

40μg/mL, 80μg/mL, 120μg/mL, 160μg/mL) was introduced into the medium using sterile forceps. Later, this setup was subjected to incubation for 24h at 37°C. Sterile broth was maintained as negative control. Control for biofilm was maintained with pathogen in the growth medium. After incubation, the catheters were removed from broth and transferred into sterile PBS phosphate buffered saline to get rid of planktonic cells and then the catheter was stained with 0.1% crystal violet (CV) for 10 mins. The catheters were dried and observed under compound microscope.

Staining solutions were made out by mixing 0.05mL of stock solution of 1% Acridine orange with 5mL of acetate buffer 0.2M (pH4). Sterile catheter was placed with AgNPs treated and untreated bacterial pathogen and allowed to dried at 50°C, the bacterial cells adhered to catheter surface was fixed with absolute methanol and stained with Acridine orange for 1 min, rinsed with distilled water and dried. The catheters were observed for fluorescence microscope [25]. The biofilm can be observed on the surface of the catheter [26].

Biofilm inhibition percentage of the urinary catheter coated with AgNPs was studied using microtiter well plate method. 50μL of TSB diluted with 10μL of mid-log phase culture was added to the wells. Different concentration of AgNPs coated catheter (20μg/mL,40μg/mL,80 μg/mL,120μg/mL,160μg/mL) was added to the respective wells. Test culture with uncoated sterile catheter act as a negative control. The well plates were incubated for 24h at 37°C [3]. After 24 h incubation, the catheters were removed and washed twice with sterile distilled water to remove the planktonic cells. Catheter containing biofilm was stained with 1mL of 0.4% CV solution and then washed with sterile distilled water to remove excess stain. Stain was then solubilized by 1mL of absolute ethanol. The well plates were read for OD value at 590nm using micro titer plate reader. Conducted experiments were done in triplicate and graph was drawn using graph pad prism version 9.1.2.

The inhibition percentage was calculated by the formula

$$\frac{(Ab_c - Ab_t)}{Ab_c} \times 100 \tag{2}$$

Where $Ab_c$ = absorbance of control well $Ab_t$ = absorbance of test well.

## Antibacterial activity of AgNPs coated urinary catheter

Antibacterial activity of AgNPs coated catheter was assessed by the following procedure. Each test bacterial strain of 0.5 McFarland standards [19,27] was swabbed on MHA plates using a sterile swab. AgNPs coated catheter and uncoated catheter was situated on agar and incubated at 37 ºC for 24 h and zone of inhibition was observed and measured [27].

## SEM analysis of urinary catheter

AgNPs coated catheter and uncoated catheter pieces were introduced into trypticase soy broth which is inoculated with a strong biofilm former *E. coli* AMB4, aseptically for 48 h at 37 ºC. To analyze SEM, catheters were fixed with 2.5% of Glutaraldehyde in 0.1M sodium phosphate buffer for 3 hours and washed with 0.1M sodium phosphate buffer. Then the sample was allowed to dehydrate through a series of ethanol wash: 30%, 50%, 80% for 10 min [21,28].

# Result

## Optimization of AgNPs production

Initially the preliminary confirmation of AgNPs production in the reaction mixture through green process was observed through the visual color change followed by surface plasmon

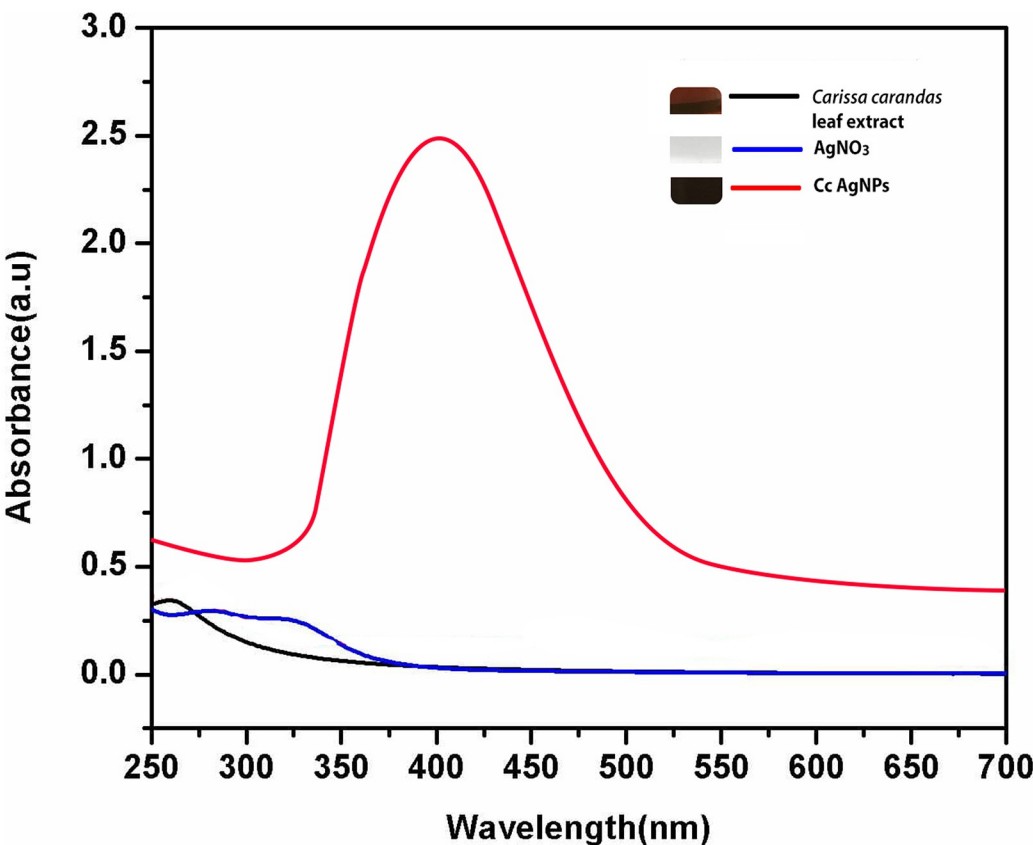

**Fig 1. UV-visible spectra of *Carissa carandas* leaf mediated synthesized AgNPs (before optimization procedure at pH7 with 1mL of *C. carandas* leaf extract, 1mM of AgNO$_3$ and a 30 mins reaction), AgNO$_3$, *C. carandas* leaf extract.**

resonance (SPR) using UV–visible spectroscopy as a tremendous tool. An intense peak at 410 nm by UV–visible absorption spectra confirmed the formation of colloidal AgNPs. *Carissa carandas* leaf extract pH was found to be pH 7 and the UV spectra of the leaf extract was observed as shown in the Fig 1. There is no interesting $\lambda_{max}$ peak in *C. carandas* leaf extract and silver nitrate solution as shown in the Fig 1. Optimum reduction of Ag$^+$ by *C. carandas* leaf extract to attain the maximum AgNPs production was succeeded by modifying the pH, substrate concentration, silver ion concentration, and production time and their wavelength were revealed in Fig (2A)–(2D). In summary, pH is one of the most important variables in nanoparticle products. In acidic environment, particles did not form (pH 2 and 3). At alkaline pH 10, the color production occurred quick, although only weak peak was visible. The reaction was begun as soon as the silver nitrate was introduced to the reaction at neutral pH 7. The solution changed color from pale yellowish to dark brown, indicating the production of silver nanoparticles. Production of AgNPs was further verified by the characteristic absorption peak (Fig 2A) at 410 nm in the UV-visible spectrum. Interestingly a strong intense peak was observed at pH 9 at the same wavelength of 410 nm, however, the width of the absorption peak corresponding to the SPR of AgNPs was increased, indicating the agglomeration of the AgNPs.

Different concentration of *C. carandas* leaf extract was optimized for maximum production of AgNPs. However, the different extract concentration shows peak at 410 nm.

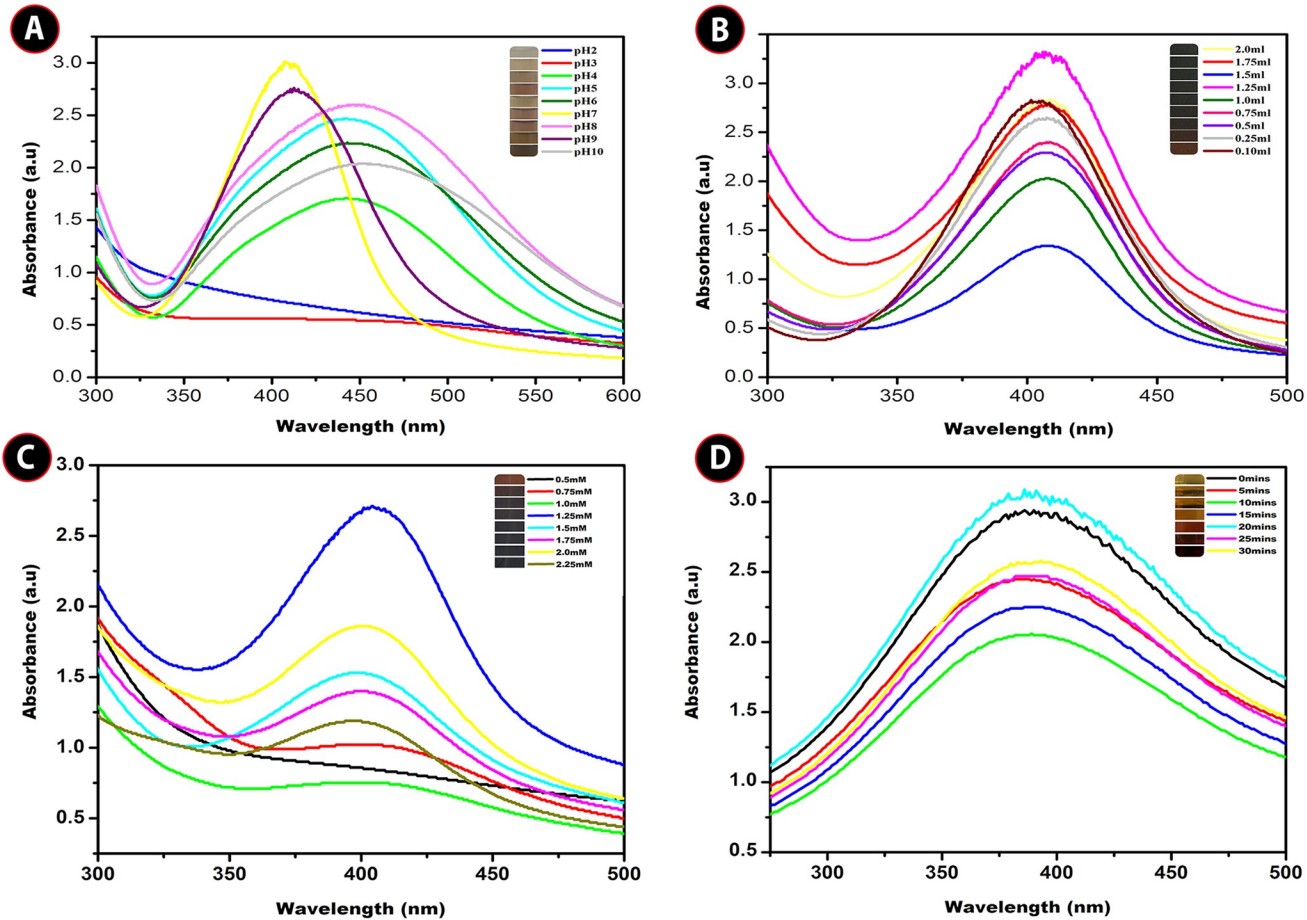

**Fig 2. UV—Vis spectra of AgNPs suspension obtained with *Carissa carandas* leaf extract at (A) different pH (B) different substrate concentration (C) different silver ion concentration (D) different time intervals.**

Interestingly 10mL of reaction mixture containing 1.25 mL of leaf extract (Fig 2B) was turned to dark brown immediately after the addition to 0.1mM of silver nitrate solution at an optimized pH 7.

Different concentration of silver nitrate was optimized for the maximum synthesis of AgNPs. 1.25 mM concentration of silver nitrate (Fig 2C) shows a strong intense peak at 410nm and the reaction mixture was turned immediately to dark brown after the addition optimized leaf extract of 1.25 mL and altering to optimized pH 7. However, 2.0 mM, 1.75 mM and 1.5mM silver nitrate concentration shows much weaker absorbance peak at 410 nm.

Time taken for the maximum AgNPs production was optimized by measuring the reaction solution in UV-visible spectroscopy at a various time interval, where the reaction mixture contains optimized silver nitrate concentration of 1.25 mM with optimized substrate concentration of 1.25mL at an optimized pH 7. And the dark brown color occurred within 20 min of incubation, suggesting that AgNPs formed quickly. However, the color change observed in 25 and 30 mins was very dark than the color obtained in 20 mins (Fig 2D), the absorbance spectra at 25 and 30 mins showed weak characteristic peak. As a result, the optimized medium enabled for the greatest production of silver nanoparticles, and the reaction took place quickly.

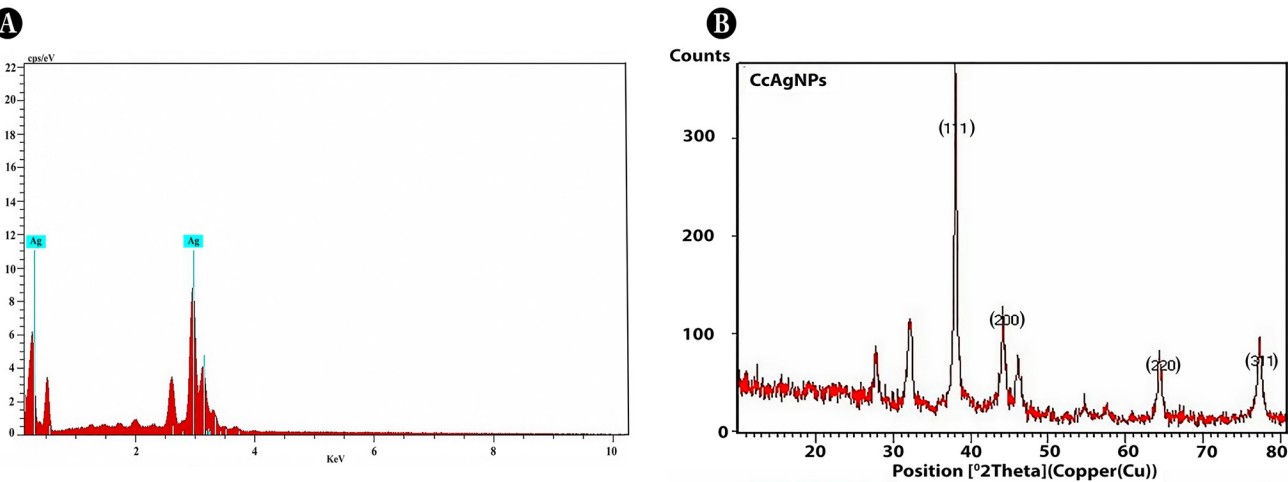

**Fig 3. Characterization of AgNPs synthesized using *Carissa carandas* leaf extract using (A)EDX (B) XRD.**

## Characterization of nanoparticles

**EDS.** The presence of silver element in synthesized AgNPs was confirmed by Energy Dispersive analysis Fig 3(A). The metallic AgNPs shows a typical optical absorption peak at 3KeV. The peaks of silver element were obtained at 3keV from the synthesized AgNPs particle and some weaker peaks were also observed that corresponds to oxygen (O) and carbon (C) element.

**XRD.** XRD pattern was evaluated to resolve the width, peak position and peak intensity in 2θ spectrum ranging from 20º to 80º as depicted in Fig 3(B). The characteristic peaks at 38.01, 44.13, 64.46, 77.40; Bragg reflections corresponding to [111], [200], [220] and [311] lattice plans of FCC structure (JCPDS File No. 04–0783) of AgNPs were observed. This pattern shows the crystalline structure of AgNPs, size of AgNPs was calculated by full width at half-maximum (FWHM) data with the Scherrer formula $D = K \lambda/\beta \cos\theta$ was estimated to be 25.4 nm. Where k = constant, λ = X-ray wavelength, β = angular FWHM, θ = Bragg's diffraction angle and D = crystalline size of diffraction angle θ.

In addition, three unassigned peaks appeared at 27.99º, 32.13º and 46.28º. These peaks were weaker than those of silver. This may be due to the bioorganic compounds occurring on the surface of AgNPs. The appearances of these peaks are due to the presence of phytochemical compounds in the leaf extracts. The stronger planes indicate silver as a major constituent in the biosynthesis.

**FTIR.** The FTIR spectrum of AgNPs shows major absorption band around 440.02, 479.57, 548.00, 1104.68, 1383.22, 1443.38, 1621.55, 2921.60, 3419.99cm$^{-1}$ and the crude *C. carandas* leaf extract shows absorption spectra on 780.44, 1105.57, 1315.55, 1386.44, 1443.56, 1617.79, 2922.97, 3421.32cm$^{-1}$ depicted in Fig 4(D). The peak on 440.02 was due to aryl disulphide stretches, 479.57cm$^{-1}$ was due to polysulphide stretches, 548 due to C-I stretches and 1104.68 and 1105.57 were–C-O- stretching vibration of alcohol and phenol, 1443.38 and 1443.56cm$^{-1}$ were–C = C- aromatic structures, 1621.55and 1617.79 were the -C = C- alkene group. Peaks 2921.60, and 2922.97cm$^{-1}$ were -cHsp3 group and the band on 3419.99 and 3421.32cm$^{-1}$ were the normal polymeric stretch of hydroxyl (OH) group. The absorption band is due to the vibration effect of the alkaloids, terpenoids and flavonoids present in the plant extract and plays crucial role in capping and stabilization of AgNPs. The band shift of hydroxyl group in the FTIR spectra confirmed the binding of Ag$^{+}$ to the OH group. All the changes in

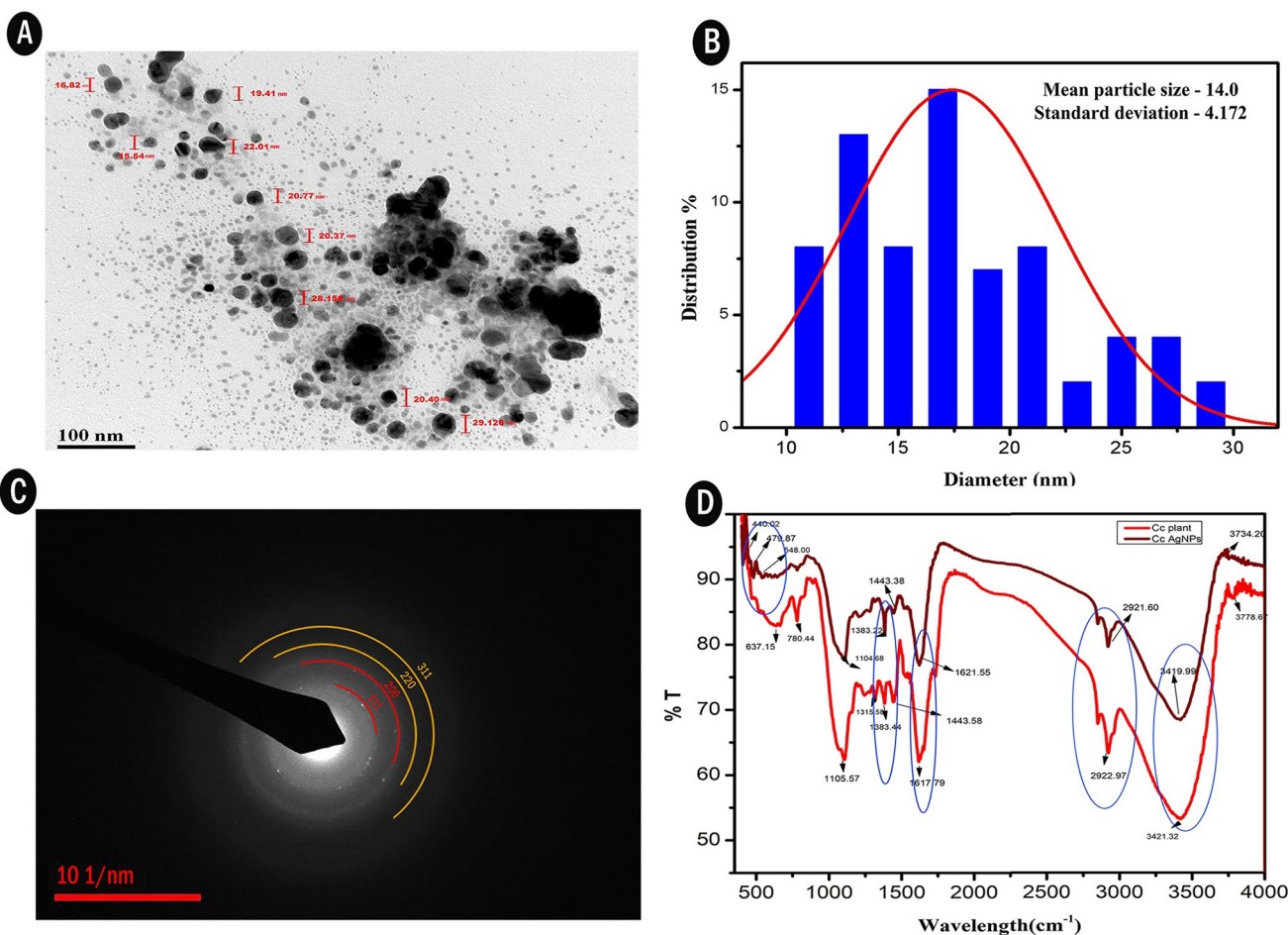

**Fig 4. Characterization of AgNPs synthesized using *Carissa carandas* leaf extract using (A)TEM (B) Histogram (C) SAED (D) FTIR.**

peak support the impact of functional group in *C. carandas* leaf extract as reducing and stabilizing agents to synthesize AgNPs. Some peaks appeared in the FTIR spectrum of leaf and disappeared in AgNPs spectrum. The disappearance of peaks suggests that phytochemical present in the extract involved in the reduction of AgNPs [29].

**HR-TEM.** High resolution Transmission electron microscope determined the morphology, shape and size of bio fabricated AgNPs as shown in the Fig 4(A). We have analyzed TEM micrograph using Image J software and from the analysis we have found the particles was polydispersed and predominantly found to be spherical with the average diameter of approximately 14nm were determined through the histogram obtained Fig 4(B). SAED pattern image of AgNPs revealed the diffraction rings from inside to outside, could be indexed as [111, 200, 220, 311] reflections respectively with some bright spots due to Bragg's reflection, corresponding to face-centered cubic (fcc) silver was depicted in Fig 4(C).

## Antibacterial activity

Antibacterial activity of synthesized AgNPs was evaluated against Gram positive and Gram negative uropathogens such as *S. aureus*, *E. coli* and *P. aeruginosa*. The clear zone was gradually

**Table 1. Antibacterial activity against uropathogens.**

| Bacteria | ZoI of *C. carandas* (mm) | | | | |
|---|---|---|---|---|---|
| | 25µg/mL | 50µg/mL | 75 µg/mL | 100µg/mL | 125 µg/mL |
| *S. aureus* | 8±0.3 | 10 ±0.3 | 13 ±0.3 | 15 ±0.3 | 17 ±0.1 |
| *E. coli* | 10 ±0.1 | 13 ±0.2 | 13 ±0.2 | 13 ±0.3 | 15 ±0.2 |
| *P. aeruginosa* | 8 ±0.2 | 9 ±0.2 | 10 ±0.5 | 13 ±0.2 | 15 ±0.1 |

increased based on the dose dependent manner as shown in the Table 1 and Fig 5. The well diffusion assay also performed for comparative study of crude extract, AgNO$_3$ solution, standard antibiotic ciprofloxacin (50µg/mL), AgNPs, DMSO as a solvent control as shown in Fig 6 and these results were depicted in the Table 2.

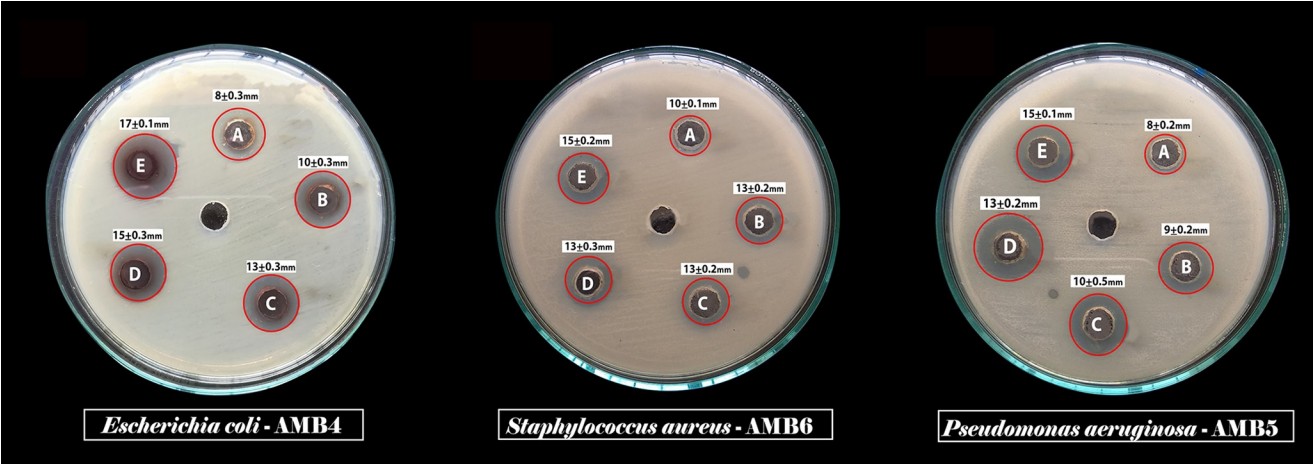

**Fig 5. Antibacterial activity of different concentrations of *C. carandas* mediated synthesized AgNPs against the test pathogens.** Zone of Inhibition in different concentrations (A-25µg/mL, B-50 µg/mL, C-75 µg/mL, D-100 µg/mL, E-125 µg/mL) of AgNPs against *Escherichia coli* AMB4, *Staphylococcus aureus* AMB6, *Pseudomonas aeruginosa* AMB5 are shown.

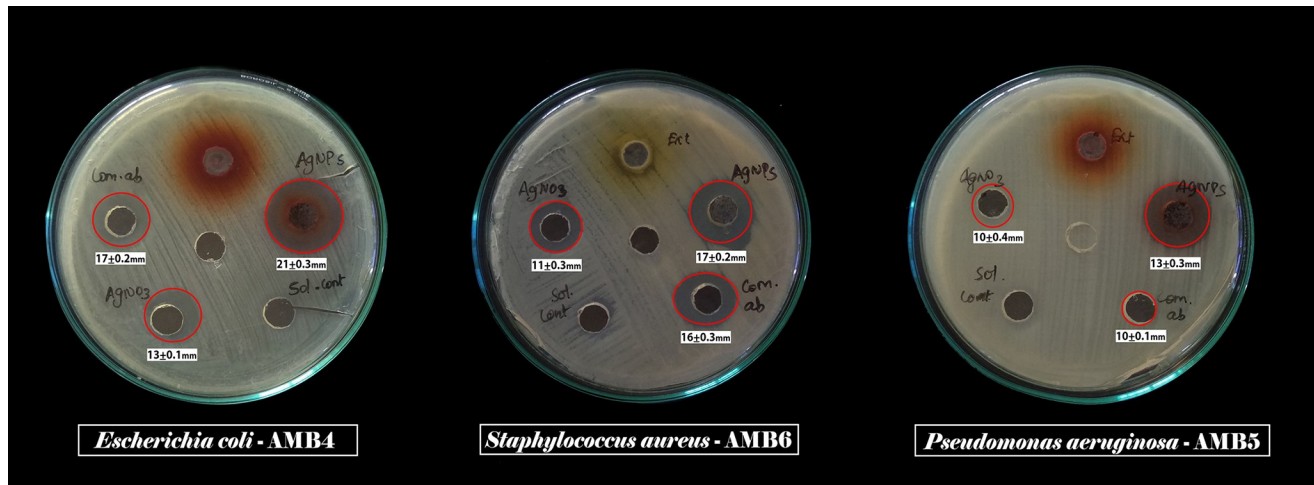

**Fig 6. Antibacterial comparison of *C. carandas* mediated synthesized AgNPs, commercial antibiotics (ciprofloxacin), *C. carandas* leaf extract, AgNO3 against test pathogens.** Zone of inhibition observed in the well of AgNPs, solvent control (DMSO), AgNO3 and commercial antibiotic (ciprofloxacin) against *Escherichia coli* AMB4, *Staphylococcus aureus* AMB6, *Pseudomonas aeruginosa* AMB5 are shown.

**Table 2. Comparative analysis against uropathogens (ZOI).**

| Strains | ZoI of *C. carandas* (mm) | | | | | |
|---|---|---|---|---|---|---|
| | Crude extract | AgNO₃ | AgNPs | Ciprofloxacin | Uncoated catheter | AgNPs Coated Catheter |
| *S. aureus* | - | 11 ±0.3 | 17±0.2 | 16±0.3 | No zone | 17±0.4 |
| *E. coli* | - | 13±0.1 | 21±0.3 | 17±0.2 | No zone | 21±0.3 |
| *P. aeruginosa* | - | 10±0.4 | 13±0.3 | 10±0.1 | No zone | 13±0.1 |

## Minimum Inhibitory (MIC) and Minimum Bactericidal Concentration (MBC)

After 24 h of incubation at 37˚C, turbidity was noticed in the *E. coli* AMB4 well plates 20 and 40 μg/mL containing silver nanoparticles indicating the growth of bacteria. Whereas in the concentrations of 60, 80, 100, 120, 140, 160 μg/mL, no turbidity was seen, indicating the inhibition of bacterial growth (Fig 7). The highest concentration 160 μg/mL of AgNPs, $OD_{590nm}$ (0.18) shows 99% inhibition, whereas the minimum inhibitory concentration was found to be 60 μg/mL, $OD_{590nm}$ (0.63) shows 97% inhibition towards *E. coli* AMB4. The MHA plates also

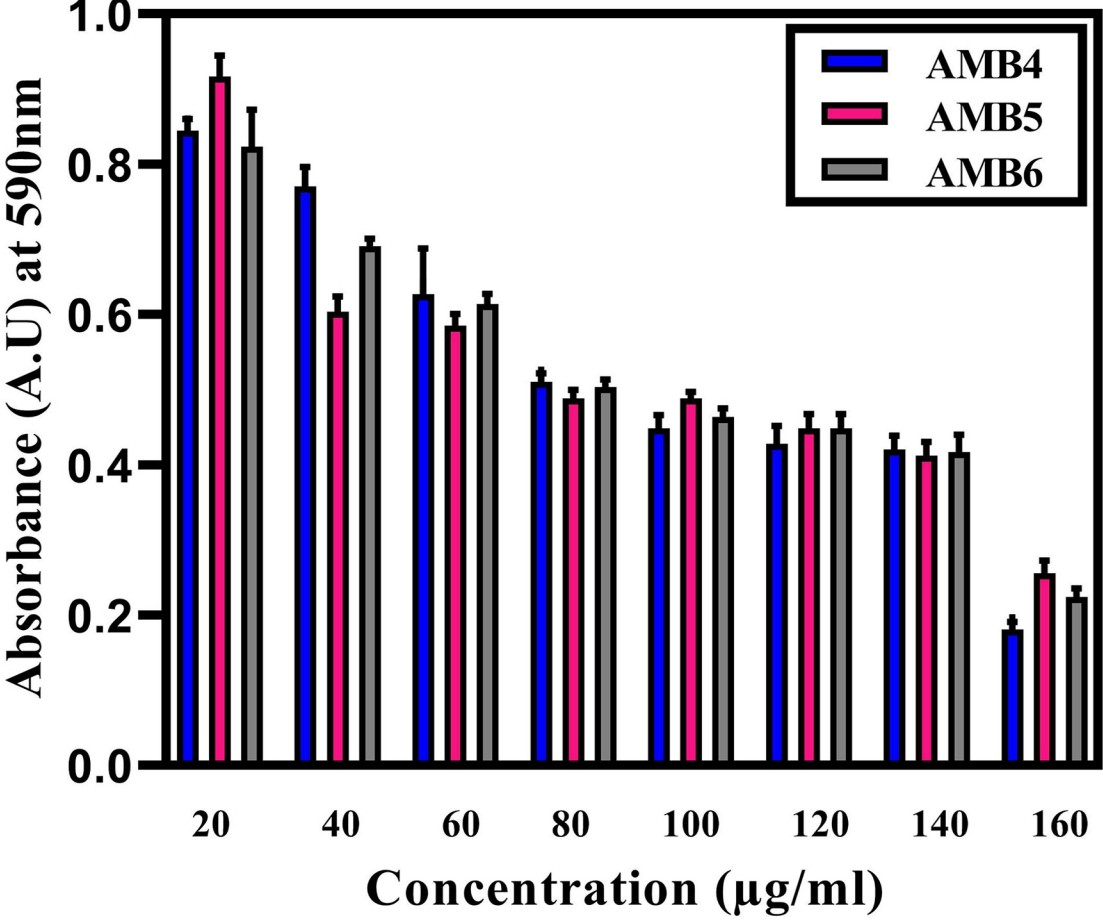

**Fig 7. Minimum inhibitory concentration for different concentrations (20, 40, 60, 80, 100, 120, 140, and 160μg/ml) of AgNPs against *Escherichia coli* AMB4, *Pseudomonas aeruginosa* AMB5, *Staphylococcus aureus* AMB6.**

**Table 3. Zone of inhibition of different antibiotics against uropathogens with presence and absence of AgNPs.**

| Strains | Antibiotics ZoI (mm) | | | |
|---|---|---|---|---|
| | | Antibiotics (a) | Antibiotic + AgNPs (b) | Increase in fold area ($b^2-a^2/a^2$) |
| *S. aureus* | Ciprofloxacin | 16 | 19 | 0.41 |
| | Gentamycin | 27 | 29 | 0.15 |
| | Trimethoprim | No zone | 11 | 2.3 |
| *E. coli* | Ciprofloxacin | 22 | 24 | 0.19 |
| | Gentamycin | 19 | 20 | 0.10 |
| | Trimethoprim | 7 | 10 | 1.04 |
| *P. aeruginosa* | Ciprofloxacin | 23 | 25 | 0.18 |
| | Gentamycin | 16 | 19 | 0.41 |
| | Trimethoprim | 5 | 11 | 3.84 |

show no bacterial growth from the concentrations of 60, 80, 100, 120, 140, 160 μg/mL, hence confirming it as bactericidal.

Similarly, *S. aureus* AMB6 and *P. aeruginosa* AMB5 well plate containing AgNPs showed turbidity in 20 μg/mL, whereas no turbidity was seen in the concentrations of 40, 60, 80, 100, 120, 140, 160 μg/mL containing AgNPs indicating the bacterial inhibition (Fig 7). The highest concentration 160 μg/mL of AgNPs, $OD_{590nm}$ (0.22) shows 99% inhibition for *S. aureus* AMB6 and highest concentration 160 μg/mL of AgNPs, $OD_{590nm}$ (0.25) shows 99.5% inhibition for *Pseudomonas aeruginosa* AMB5. Therefore, MIC of *S. aureus* AMB6 was found to be 40 μg/mL with $OD_{590nm}$ (0.69) shows 97% inhibition and MIC of *P. aeruginosa* AMB5 was found to be 40 μg/mL, $OD_{590nm}$ (0.60) shows 97% inhibition. The MHA plates also show no bacterial growth from the concentrations of 40, 60, 80, 100, 120, 140, 160 μg/mL, hence confirming it as bactericidal.

## Synergistic effect of silver nanoparticles with commercial antibiotics

In the present work, 3 commercial antibiotics were tested alone and with AgNPs against the test pathogens. AgNPs alone showed antimicrobial activity and commercial antibiotics also showed antimicrobial activity when the AgNPs is combined with the commercial antibiotics, the antimicrobial activity increased with increased fold as it was evidenced in Table 3. Maximum increase in fold area was 3.84 and 2.3 against trimethoprim (Table 3). The synergistic antimicrobial activity against *P. aeruginosa* was better than that of *E. coli* and *S. aureus*. Maximum increase in fold was 3.84 against trimethoprim 1.04 for *E. coli* while it was 2.3 for *S. aureus* against trimethoprim (Table 3).

## Bacterial biofilm potential

In our study, the biofilm forming ability was verified by test tube method. The test tube base contains the adhered layer of uropathogens. *P. aeruginosa* forms a strong biofilm mat than another organism. The biofilms were analyzed quantitatively to check the potential biofilm formers, *P. aeruginosa* shows $OD_C$ (0.1784) < OD (3.045) however *S. aureus* also produce strongly adherent biofilm layer $OD_C$ (0.1784) < OD (3.1074), *E. coli* shows an $OD_C$ (0.1784) < OD (3.012) confirms that it is a strong biofilm former.

## Biofilm inhibition in AgNPs coated catheter

AgNPs coated catheter (Fig 8) was evaluated for the anti-biofilm activity against the uropathogens. Uropathogens adhered to the surface of catheter was treated with different concentration

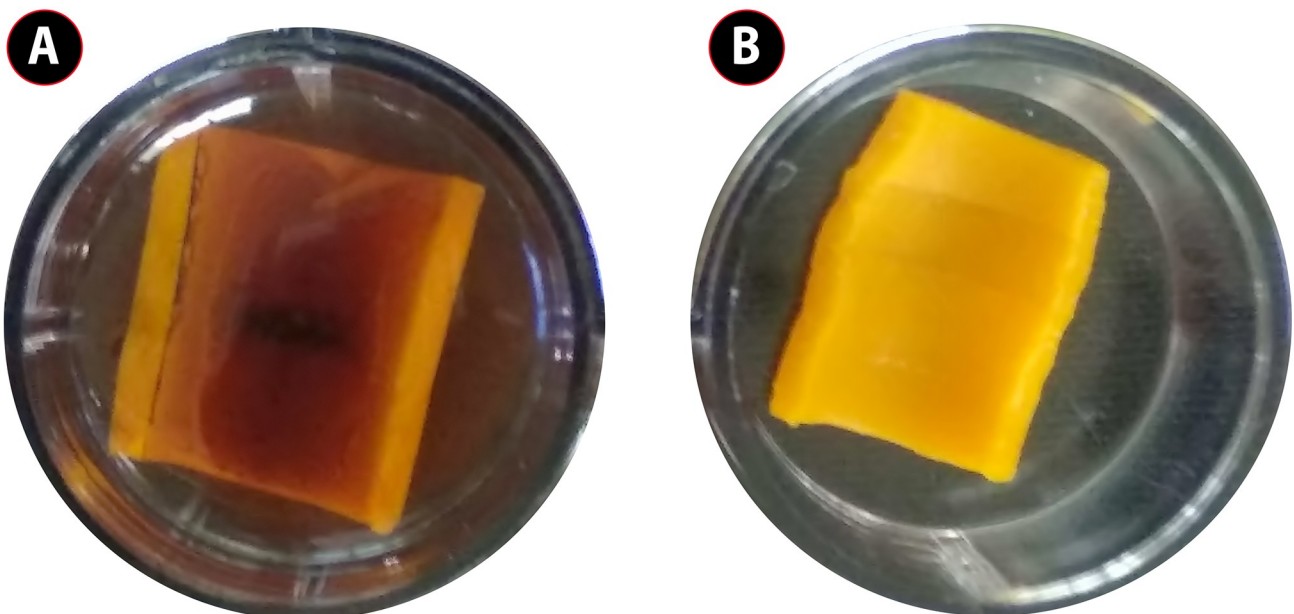

**Fig 8. Urinary catheter coated with AgNPs and uncoated catheter (A)** *C. carandas* **leaf mediated synthesized AgNPs coated urinary catheter of size 1× 1 cm (B) uncoated urinary catheter of size 1×1 cm.**

of AgNPs and subjected to microscopic analysis. Under the microscopic observation tightly adhered cells are gradually dispersed depending upon the concentration of NPs compare whereas control showed an adhered mat formation as shown in S1 Fig. Viability and disruption of biofilm mat after AgNPs treatment was analyzed by fluorescence microscopy, showed an abruption of biofilm on AgNPs coated catheters as shown in S2 Fig. The dense biofilm mat on uncoated catheter using an acridine orange staining method. In quantitative assay, highest concentration of AgNPs coated catheter showed the highest level of inhibition. The inhibition of *Pseudomonas aeruginosa* 85.8 ± 1.450% was slightly higher than the *S. aureus* 82.8 ± 1.83% whereas the inhibition percentage of *E. coli* 71.4 ± 1.25% become lesser than the other two test pathogen. Percentage of inhibition was calculated and shown in Fig 9.

### Antibacterial activity of AgNPs coated urinary catheter

Antibacterial activity of AgNPs coated urinary catheter and uncoated catheter as shown in the Fig 8 was evaluated where 40μg/mL of AgNPs coated catheter exhibits antibacterial activity with the value of 17±0.4, 21±0.3, and 13±0.1 for *S. aureus* AMB6, *E. coli* AMB4, and *P. aeruginosa* AMB 5, respectively. The urinary catheter impregnated with AgNPs shows ZOI against uropathogens whereas uncoated catheter shows no zone of inhibition (Table 2).

### SEM analysis of urinary catheter

SEM analysis of AgNPs coated catheter Fig 10(A) clearly shows the strong overlaying of AgNPs on the catheter surface and uncoated catheter Fig 10(B) shows a clear image of catheter surface. Further, SEM imaging was done on the AgNPs coated catheter inoculated with strong biofilm former *E. coli* AMB4 Fig 10(D) states the biofilm mat formed by the *E. coli* AMB4 was disturbed due to the activity of AgNPs and Fig 10(C) clearly shows the dense biofilm mat on the surface of the uncoated catheter inoculated with *E. coli* AMB4 which proves that *E. coli*

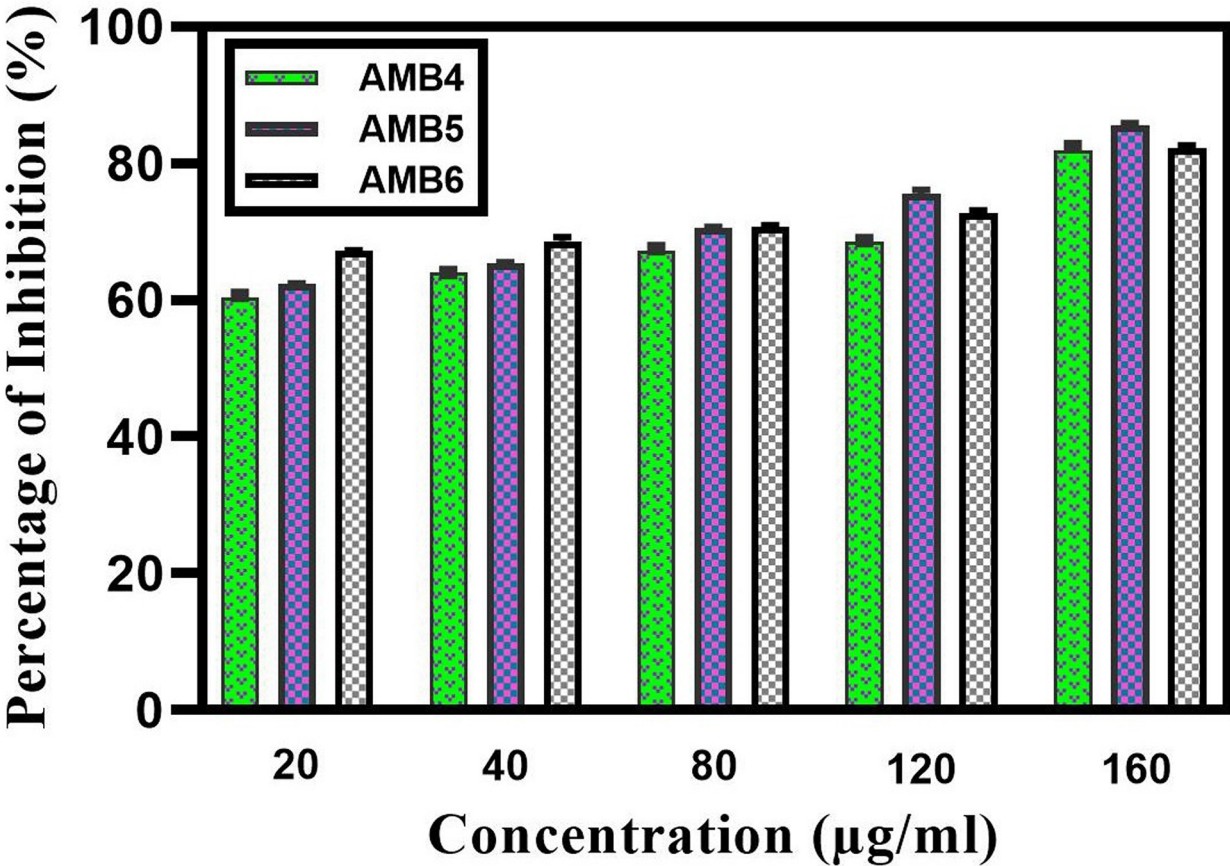

**Fig 9. Biofilm inhibition percentage of AgNPs coated catheter.** AgNPs coated catheter with different concentration of 20,40,80,120,160 μg/mL shows biofilm inhibition towards *Escherichia coli* AMB4, *Pseudomonas aeruginosa* AMB5, *Staphylococcus aureus* AMB6.

AMB4 is a strong biofilm former. The incorporation of urinary catheter (biomedical devices) with AgNPs provides better biocompatibility.

## Discussions

Uropathogens are the major cause of UTI with their biofilm formation. These uropathogens are notorious and perpetuating. They become combat against wide range of antibiotics and environmental stress such as host immune response. They are difficult to treat and eradicate [30]. The major toughness of biofilm is architecture EPS, quorum sensing (QS) activity. The over production of EPS leads to resistant against antibiotic and another crucial factor is QS (construction of wild type architecture) it increases the stability against oxidative and osmotic stresses of biocide [31] Milan et al. [32] states that nosocomial acquired UTI shows high level of resistant than community acquired UTI show the patient indwelling catheters shows high risk of UTI. Due to its biocompatibility and backdrop of antimicrobial resistant create the thirst of seeking naive therapeutic despite of antibiotic [33]. The plant derived drug compiled with nanotechnology wrap out the resistance against Uropathogens. In this present study, *C. carandas* leaf extract was subjected to synthesize silver nanoparticle, with potent antibacterial and antibiofilm activity. The choice of green synthesis of NPs was due to their capping capability and stability. Biosynthesized NPs are facile; cost of effective, fast, non-toxic, possessing well defined morphology and uniformity in size [34]. $Ag^+$ capped with the phytomolecules present in the plant

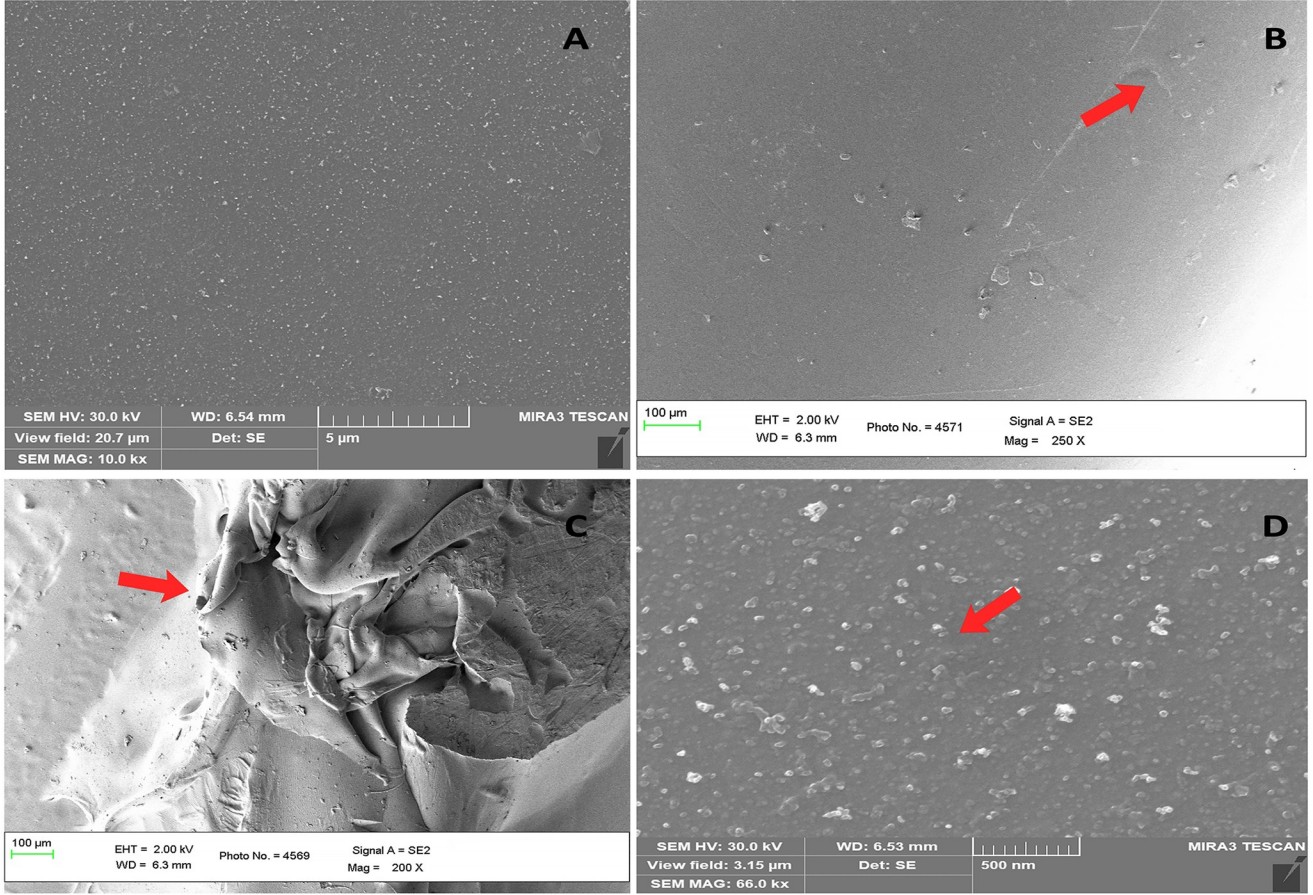

**Fig 10. SEM analysis of urinary catheter (A) SEM micrograph of uncoated urinary catheter (control) (B) SEM micrograph of urinary catheter coated with 30 μg/mL of AgNPs, arrow indicate the coating of AgNPs (C) SEM micrograph of biofilm mat formed by *Escherichia coli* AMB4 over uncoated urinary catheter, arrow indicates the mat formation (D) SEM micrograph showing the disruption of biofilm formed by *Escherichia coli* AMB4 over AgNPs coated urinary catheter, arrow indicates the disruption of biofilm.**

enhanced the antimicrobial activity. Fig (2A)–(2D) demonstrates the absorption spectra of SPR for the optimization of AgNPs synthesis under distinct parameters viz. pH, crude extract concentration, Ag ion concentration and incubation time for analysis. These results provide for evaluating the reaction parameter and optimized conditions for NPs synthesis [35] Ibrahim [36] stated that, reaction mixture color and SPR intensity which are pH dependent.

In our study, acidic and alkaline pH shows weak absorbance peak. However, strong intense peak was observed in pH 9, agglomeration of reaction was happened. The neutral pH 7 typically increased the absorbance peak and provide a favorable environment. Crude concentration is noteworthy due to their phytochemical stabilizing agents. The raising of absorption peak was noticed in in 1.25 mL of extract concentration. Whereas the addition of higher crude concentration leads to decreased absorbance peak [37]. The absorption peaks were gradually increased with the increased metal concentration which may be attributed by longitudinal vibrations [38]. Optimized parameters for maximum biosynthesis of AgNPs was established at 1.25mM of AgNO$_3$, 1.25mL of substrate, pH7 and a time reaction of 20 minutes. The color change of the heterogeneous reaction mixture observed at 410nm due to their electron excitation similar observation [39]. FTIR peak of our study was in accordance to Pavia et al. 2009 [40], the peaks ranging from 3200–3600 cm$^{-1}$ are related to the O-H and -NH$_2$ stretching

vibrations and suggest that hydroxyl and carbonyl groups may responsible for the synthesis and stabilization of AgNPs [41], the peak at 2921.60 and 2922.97 are assigned to C-H stretching [40]. According to Mariselvam et al. [42] absorption band ranging from 1700–1600 cm$^{-1}$ in the spectra confirms the formation of AgNPs. The bands observed at 1383.22 cm$^{-1}$ and 1386.44 cm$^{-1}$ corresponds to the C-N stretching vibration of aromatic amine [43]. The presence of amines or alcohols or phenols represents the polyphenols capped by AgNPs [44,45]. The shifting peak up and down reveals the synthesis of AgNPs. Biomolecules in *C. carandas* leaf extract is responsible for the stabilization of AgNPs [46]. The FTIR analysis speaks the stretch band and bond of AgNPs, the presence of potential biomolecules with Ag attachment leads stabilization and capping [3,19]. Due to their surface adhered potential biomolecules, green mediated AgNPs shows the higher anti-bacterial and anti-biofilm activity [47]. The size and shape of AgNPs plays a major role in bactericidal activity [48]. XRD analysis revealed the crystalline nature of AgNPs presence of silver confirmed by the diffraction pattern. These XRD patterns reported in earlier studies Saratale et al. [49] was accordance with our results. EDX profile outcomes exhibits the strong signal for silver approximately at 3KeV due to the SPR which is identical to Ramar et al. [50] and Magudapathy et al. [51] for the production of leaf extract mediated synthesis AgNPs. The structure and size of NPs were concluded as spherical and polydispersed with the approximate size of 14nm was confirmed by HR-TEM analysis [52]. SAED pattern of AgNPs was shown in the Fig 4C. Further ring like diffraction pattern indicates that the particles are crystalline [53]. During recent years, undesirable consequence effect of catheter related UTI infections lead to the increased mortality [54]. Application of AgNPs shows the efficient antimicrobial activity and that are justifiable tool for evading indwelling catheter related infections. Medically implantable devices coated with AgNPs which are requisite factor for evading the bacterial adherence and agglomeration of biofilm [55] in this investigation reported that, *E. coli* (71.4%)), *S. aureus* (82.8%), *P. aeruginosa* (85.8%) these nosocomial clinical pathogens are prevalent in formation of biofilm. These results were similar to Sharma et al. [56] and Kamarudheen and Rao [57]. The AgNPs embedded catheter shows antimicrobial activity against uropathogens which may due to their size and inhibition capacity that makes the drug resistant uropathogens susceptible [58]. The commercial catheters coated with AgNPs (Fig 8) creates the efficiency against the UTI. Urinary catheters are the major cause of biofilm formation in urinary tract results in nosocomial infection [59]. Techniques followed to coat urinary catheter as layer by layer for enzyme coating, impregnation of antimicrobial agents [60], polycationic nanosphere coating [61], impregnation of complex molecules [62]. In recent years, impregnation of urinary catheter with silver is under practice [63]. AgNPs is a fast and promising strategy for bactericidal coating on silicone based medical devices [64]. In recent years, there is rise in mortality rate associated with catheter associated urinary tract infection [65]. Therefore, it is important to coat the medical devices with antimicrobial agents. AgNPs are excellent tool for avoiding catheter associated UTI [55]. The solid surface provides a strong anchoring habitation for bacteria to form biofilm, similarly biofilm is formed on the surface of implant device, which protects the bacteria from antibiotic action and cause several infections [66]. Additionally, functionalized, immobilized and surface modified AgNPs embedded on surface of implants are inhibiting bacterial adhesion and *ica*AD transcription in implants [67].

AgNPs are responsible for the anti-cancer, anti-oxidant, anti-microbial activity [68]. The AgNPs reduces the encrustation of obstinate biofilm and ruptures and disintegrate the biofilm mat and shows bactericidal activity against uropathogens. The coated catheter shows antibacterial, anti-EPS and anti-quorum sensing activity of uropathogens and end up the pathogens into avirulent and disrupt the biofilm [69]. Studies shows that AgNPs has the ability to destroy the biofilm structure and it was further evidenced by acridine orange staining method [70].

Fluorescence microscopy (S2 Fig) shows the bacterial biofilm formation over uncoated urinary catheter by uropathogens whereas biofilm disruption was observed in the AgNPs coated urinary catheter exposed to uropathogens. This biofilm disruption can occur due to the inhibition of bacterial growth and adhesion to the surface [71]. The AgNPs coated biomaterials has a lethal impact on bacterial cells, which could be observed as shrunken cells when compared to live cells in fluorescence microscopy [72]. The unrestricted growth of untreated control cells resulted in the development of abundant biofilm on the surface as seen in [S2a, S2c and S2e Fig]. Treatment with *Carissa carandas* AgNPs, on the other hand, reduced biofilm development as seen in [S2b, S2d and S2f Fig]. Similar results were observed in Mujeeb et al. 2020 [73]. Antibiofilm activity of the AgNPs coated urinary catheter was higher in *Pseudomonas aeruginosa* [74] and exhibit 86% biofilm inhibition. The *in-vitro* studies show efficient activity against uropathogens by using AgNPs coated catheters. Scanning Electron Microscopy (Fig 10) was employed to identify the biofilm formation and destruction in surface modified and unmodified catheters using AgNPs exposed to uropathogens. SEM observation clearly indicate the disruption of *E coli* AMB4 biofilms (Fig 10D). Similar results were observed in the antibiofilm activity of AgNPs by Kostenko et al. 2010 [75] and Ansari et al. 2014 [76]. SEM micrograph shows dense mat on surface of the urinary catheter (Fig 10C) formed by *E. coli* AMB4. Our results are in agreement with the study conducted by Gomes et al. a greater quantity of biofilm was produced on the hydrophobic silicone surface [77].

The AgNPs have tremendous advantage for biological applications over the bulk metal owing to it size that enables the NPs to facilitate to anchor in to the micro cell (bacteria) components [78]. AgNPs causes physical damage to the cell components leads to killing of bacteria [79] (Fig 11). Because of the cell wall, architecture, thickness varies, AgNPs antibacterial action is associated with gram positive and gram-negative bacteria [79]. Plenty of hypothesis that have been proposed, the antibacterial mechanism action has yet to be definitively established. The antibacterial mechanism (Fig 11) that we postulated based on the existing literature may be described as follows; 1) the plant mediated AgNPs adhere to the cell membrane and forms an electrostatic interaction which results in the leakage of internal substances; 2) Ag+ ions or AgNPs interact with the sulfhydryl group of enzymes and proteins [80] and inhibit the enzymatic and protein activity; 3) Cellular toxicity induced by AgNPs is triggered by reactive oxygen species (ROS) and free radicals, which destroys internal cell structures and causes cell death, lipid peroxidation, and DNA damage; 4) AgNPs interact with the ribosome and inhibit the translation process in the cell. The high surface area of AgNPs in generating silver ions explain the mechanism of AgNPs action. In the presence of oxygen and proton, aqueous AgNPs were oxidized producing silver ions when the particle dissolves [81]. The toxicity of smaller or anisotropic AgNPs with greater surface area was higher [82]. For improved antibacterial action, the greatest concentration of silver ions, quickest release of silver ions and greater surface area of silver ions are evaluated [83]. AgNPs antibacterial action is mostly owing to their capacity to generate ROS and free radical [84]. These free radicals attached to the cell wall of bacteria and generate pore, these pores ultimately cause cell death [85]. Moreover, production free radical and high levels of reactive oxygen species (ROS) are also a precise mechanism of AgNPs to inhibit bacterial by apoptosis and DNA damage [86]. There are different proposed mechanisms for antimicrobial activity of AgNPs. AgNPs (positively charged) can easily interact with negatively charged cell membrane which enhances the antibacterial activity [87]. The charges in the cell can facilitate the attraction of AgNPs for attachment on to the cell membrane [88]. AgNPs also destabilize the ribosomes and inhibit the electron transport chain [67]. AgNPs causes damages to bacteria by interfering the function of DNA replication [89], cell division and respiratory chain [90]. Because of the combination of cell wall components and AgNPs charges, the effect of AgNPs on gram positive bacteria is smaller than on gram negative

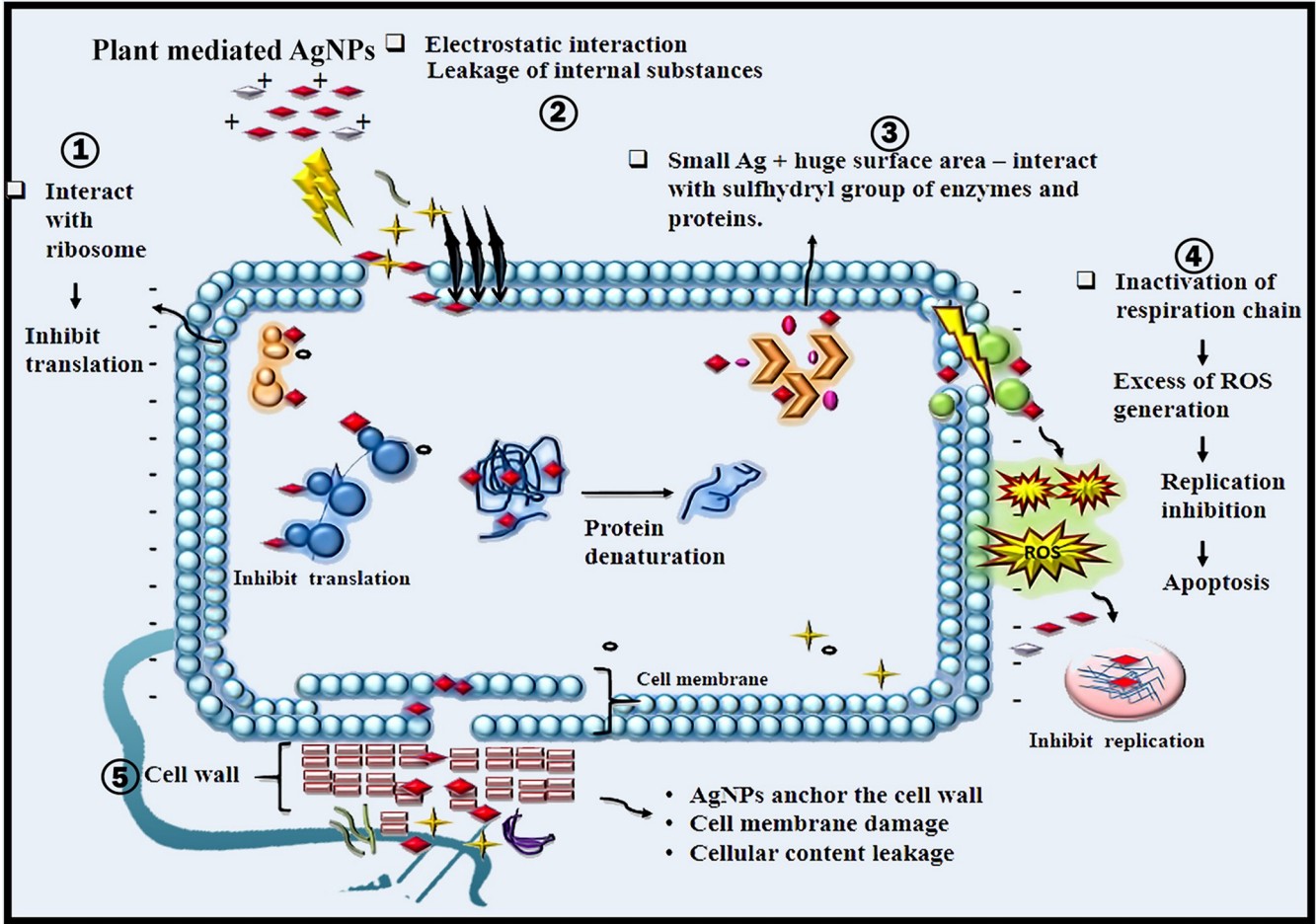

**Fig 11. Proposed antibacterial mechanism of plant mediated AgNPs showing various inhibiting properties of AgNPs.** 1) AgNPs interact with ribosome and inhibit the translation; 2) AgNPs have electrostatic interaction with the cell wall which ultimately causes the leakage of internal substances; 3) AgNPs interact with sulfhydryl group of enzymes and proteins, hence protein denaturation takes place; 4) AgNPs inactivates the respiratory chain and excess ROS generation, results in the apoptosis; 5) AgNPs anchor the cell wall of the bacteria and causes damages to the cell membrane and the cellular content get leaked.

bacteria [67]. The killing of bacteria directed through several phenomenon like penetration of AgNPs in to membrane, surface area in contact, reach cytoplasm, ribosomes, interaction with cellular structures and biomolecules by several process [81].

In this work, a mechanism for antibiofilm activity of AgNPs is proposed based on previous studies and can be summarized as follows (Fig 12): 1) AgNPs has electrostatic interaction with the cells and disturb the biofilm formation; 2) AgNPs target the eDNA to eliminate bacterial biofilm; 3) AgNPs degrade the EPS formation and breaks the biofilm mat; 4) AgNPs inhibits the signal produced by the bacteria, thereby inhibiting the biofilm formation; 5) interact with the small regulatory RNA and extracellular protein to inhibit the biofilm. The bacterial adhesion, biofilm development and biofilm integrity, as well as internal communication, are all aided by extracellular DNA (eDNA) [91]. eDNA acts as an excellent target to eliminate bacterial biofilm [92]. eDNA is polyanionic nature and electrostatic contact is mostly mediated by AgNPs that are positively charged. Through short range hydrophobic and Vander Waals force, silver ions interact with the oxygen and nitrogen atoms of DNA bases [93–95]. The electrostatic interaction, on the other hand, has an impact on cell wall thereby disturbing the formation of bacterial biofilm mat [93]. Bacterial biofilms are developed as a result of bacterial

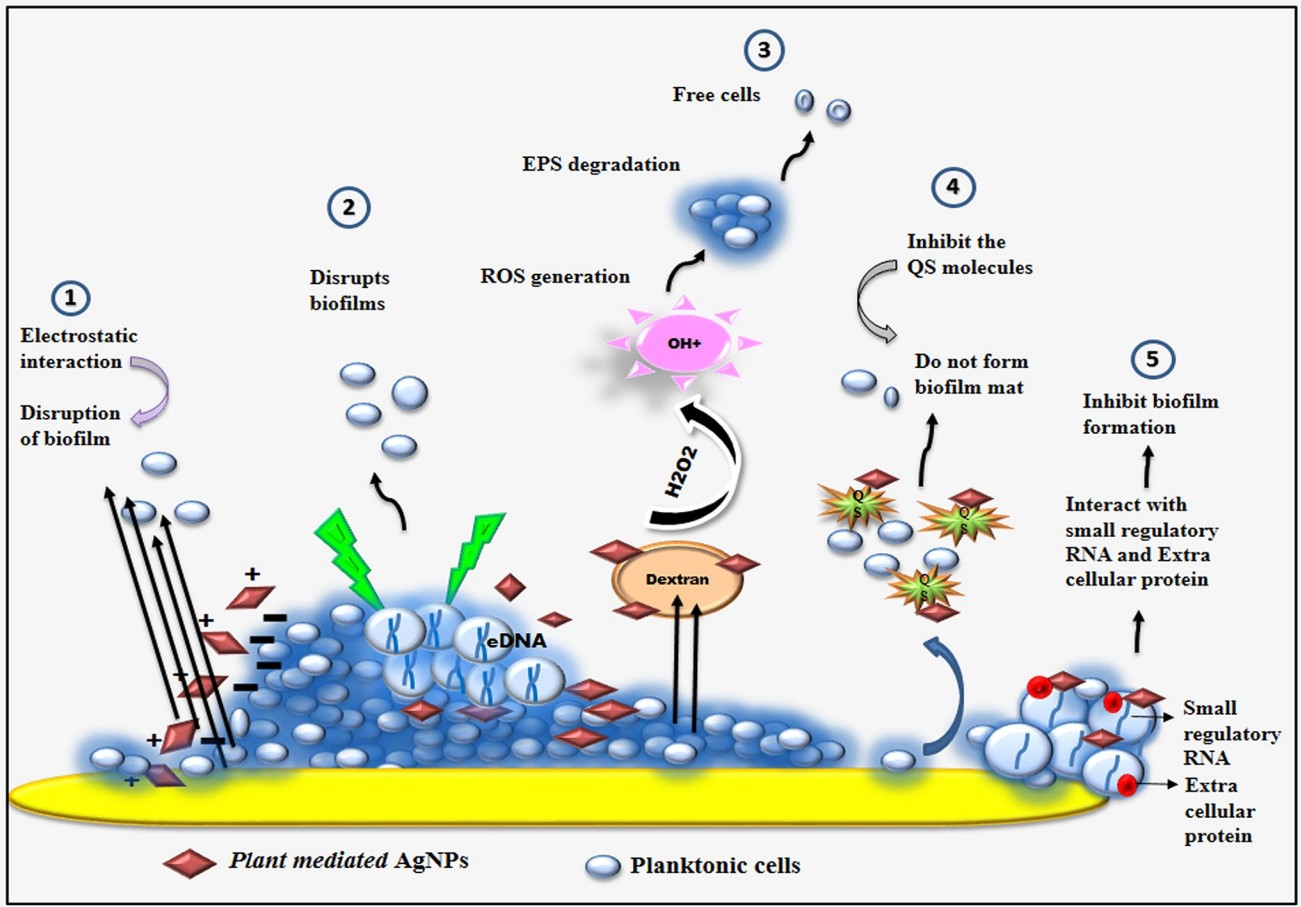

**Fig 12. Proposed antibiofilm mechanism of plant mediated AgNPs.** 1) AgNPs has electrostatic interaction with the cells and disturb the biofilm formation; 2) AgNPs target the eDNA to eliminate bacterial biofilm; 3) AgNPs degrade the EPS formation and breaks the biofilm mat; 4) AgNPs inhibits the signal produced by the bacteria, thereby inhibiting the biofilm formation; 5) interact with the small regulatory RNA and extracellular protein to inhibit the biofilm.

cells secreting EPS [96]. If the EPS production can be hindered or prevented, biofilm development will be limited as well [41]. Ansari et al. clearly state that the inhibition of EPS was primarily connected to anti- biofilm activity [97]. Several studies shows that AgNPs reduced the synthesis of extracellular polysaccharides in *P. aeruginosa* and *S. epidermidis* biofilm and their mechanism was unknown [74]. In biofilms, AgNPs interact with both cellular and extracellular RNA [98,99]. Studies shows that AgNPs interact with the small regulatory RNA, reduced biofilm and fibronectin binding by altering the RNA profile of *S. aureus* [98]. Extracellular proteins are the essential component of biofilm where AgNPs interact with these extracellular protein and extracellular polysaccharide secreted in biofilm and inhibit the biofilm formation [100]. The metallic nanoparticles bind with a few proteins that are involved in quorum sensing through hydrogen bonding, electrostatic and hydrophobic interaction makes them inactive for cell signaling [100]. Earlier, several reports on antibiofilm activity of AgNPs against several bacteria shows a promising activity [67,101,102]. Among all AgNPs interactions, AgNPs with *Pseudomonas putida* shows an innovative finding to arrest biofilm [67,101,102].

The leaf extract of *C. carandas* is said to contain a lot of flavonoids [16]. AgNPs synthesized using *C. carandas* leaf extract showed antibacterial activity [103]. The mechanism for AgNPs synthesis includes; silver ions have positive charge that attracts the functional group of

phytomolecules found in plants. The phytomolecules such as flavonoids, alcoholic and phenolic compounds, tannins, terpenoids, glycosides act as a reducing agent and reducing Ag+ ion to Ag$^o$ [104].

Hence, an overall mechanism proposed that phytochemical mediated synthesized AgNPs will open a new avenue to use as antibacterial and antibiofilm candidate after embedding in to implants.

## Conclusion

Even though, many literatures were available for silver nanoparticles, silver is gaining its attention because of its antimicrobial properties. Synthesis of AgNPs using the leaf extract will provide an ecofriendly, cheap, easily available and non-toxic. In the present study, green synthesis of AgNPs was done using *C. carandas* leaf extract, AgNPs exhibited excellent antibacterial activity towards *S. aureus* AMB6 and also showed excellent synergistic activities against *P. aeruginosa* AMB 5, AgNPs coated urinary catheter showed highest biofilm inhibition in *Pseudomonas aeruginosa* AMB5 85.8 ± 1.450%. The potential of AgNPs in inhibiting the biofilm formation supports it as a potential application for AgNPs coated medical devices. Thus, the present study helps in disclosing the biomaterial coating acts as a preventive shield against uropathogens and it is long lasting, feasible technique and it act as promising treatment for UTI and nosocomial infections.

## Supporting information

**S1 Fig. Light microscopic image of biofilm inhibition in AgNPs coated and uncoated urinary catheter.** [a, c, e] shows the biofilm formed on the surface of urinary catheter by *Escherichia coli* AMB4, *Pseudomonas aeruginosa* AMB5, *Staphylococcus aureus* AMB6 acts as control and [b, d, f] shows the biofilm inhibition activity of AgNPs coated catheter against *Escherichia coli* AMB4, *Pseudomonas aeruginosa* AMB5, *Staphylococcus aureus* AMB6.
(PDF)

**S2 Fig. Fluorescence microscopic image of biofilm inhibition in AgNPs coated and uncoated urinary catheter.** [a, c, e] shows the biofilm formed on the surface of the catheter by *Escherichia coli* AMB4, *Pseudomonas aeruginosa* AMB5, *Staphylococcus aureus* AMB6 acts as control and [b, d, f] shows the biofilm disruption by AgNPs coated catheter against *Escherichia coli* AMB4, *Pseudomonas aeruginosa* AMB5, *Staphylococcus aureus* AMB6.
(PDF)

**S1 Data. Minimal data.**
(DOCX)

## Acknowledgments

The authors thank the Vice-Chancellor and Registrar of Alagappa University for providing the research facilities.

## Author Contributions

**Data curation:** Ranjithkumar Dhandapani, Velmurugan Palanivel, Sathiamoorthi Thangavelu.

**Formal analysis:** Ranjithkumar Dhandapani, Velmurugan Palanivel, Sathiamoorthi Thangavelu, Ragul Paramasivam.

**Investigation:** Haajira Beevi Habeeb Rahuman, Ranjithkumar Dhandapani.

**Methodology:** Haajira Beevi Habeeb Rahuman, Ranjithkumar Dhandapani.

**Project administration:** Ranjithkumar Dhandapani, Sathiamoorthi Thangavelu.

**Software:** Ragul Paramasivam.

**Supervision:** Velmurugan Palanivel, Sathiamoorthi Thangavelu, Ragul Paramasivam, Saravanan Muthupandian.

**Validation:** Velmurugan Palanivel, Sathiamoorthi Thangavelu, Ragul Paramasivam, Saravanan Muthupandian.

**Visualization:** Velmurugan Palanivel, Sathiamoorthi Thangavelu, Ragul Paramasivam, Saravanan Muthupandian.

**Writing – original draft:** Haajira Beevi Habeeb Rahuman, Ranjithkumar Dhandapani.

**Writing – review & editing:** Haajira Beevi Habeeb Rahuman, Ranjithkumar Dhandapani, Velmurugan Palanivel.

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
