## [Decision Letter · Decision Letter 0]

29 Jun 2021

PONE-D-21-17789

Bioengineered phytomolecules-capped silver nanoparticles using Carissa Carandas leaf extract to embed on to urinary catheter to combat UTI pathogens

PLOS ONE

Dear Dr. Saravanan,

Thank you for submitting your manuscript to PLOS ONE. After careful consideration, we feel that it has merit but does not fully meet PLOS ONE’s publication criteria as it currently stands. Therefore, we invite you to submit a revised version of the manuscript that addresses the points raised during the review process.

We look forward to receiving your revised manuscript.

Kind regards,

Amitava Mukherjee, ME, Ph.D.

Academic Editor

PLOS ONE

Journal Requirements:

2. Please include your tables as part of your main manuscript and remove the individual files.

5. We note that Figures 8, 10 and 11 in your submission contain copyrighted images. All PLOS content is published under the Creative Commons Attribution License (CC BY 4.0), which means that the manuscript, images, and Supporting Information files will be freely available online, and any third party is permitted to access, download, copy, distribute, and use these materials in any way, even commercially, with proper attribution. For more information, see our copyright guidelines: http://journals.plos.org/plosone/s/licenses-and-copyright.

a. You may seek permission from the original copyright holder of Figures 8, 10 and 11 to publish the content specifically under the CC BY 4.0 license.

Reviewers' comments:

Reviewer's Responses to Questions

**Comments to the Author**

1. Is the manuscript technically sound, and do the data support the conclusions?

Reviewer #1: Partly

Reviewer #2: Partly

Reviewer #3: Partly

2. Has the statistical analysis been performed appropriately and rigorously? 

Reviewer #1: Yes

Reviewer #2: N/A

Reviewer #3: N/A

3. Have the authors made all data underlying the findings in their manuscript fully available?

Reviewer #1: Yes

Reviewer #2: Yes

Reviewer #3: Yes

4. Is the manuscript presented in an intelligible fashion and written in standard English?

Reviewer #1: No

Reviewer #2: No

Reviewer #3: No

5. Review Comments to the Author

Reviewer #1: 1- Throughout the manuscript, there are a lot of typical, grammatical and unclear sentences like (Scherrs formula - AgNo3- and "Among the inorganic nanoparticles Silver nanoparticles (Ag NPs) considered to be a much more attention in scientific field", that lead to misunderstanding or confusion, I recommend sending this manuscript to a professional English language editing service.

2- In the introduction, authors should justify why they decided to use Ag NPs and leaves of C. carandas? highlight their advantages, because we can not simply use something because just it is available !

3- Line 60, "Leaves of C. carandas were used to yield Ag NPs", I think you need to rephrase this sentence, as leaf extract can only be used to stabilize formed Ag NPs and / or reduce the precursor solution of silver nitrate into Ag NPs.

4- Line 93, wavelength of Cu-Kα radiation is not correct, the correct value is 1.5406 Å.

5- In line 225, authors used Scherrer formula to determine crystalline size, and they mentioned non-correct wavelength in Line 93, then accordingly, the calculated size will not be correct. Please check this size again.

6- XRD pattern contains non-assigned peaks, please explain.

7- on FTIR spectra, it is better to highlight, peaks confirming the conjugation between Ag NPs and the extract.

8- On SAED pattern, you should assign the crystalline planes and match them with those obtained by XRD.

9- Fig. 2 is not clear, it is better to draw the data using suitable software !

10- Fig. 3 it is hard to see the label, also indicate the ZOI on the figure for each tested sample.

11- Fig.4, error bars should be added.

12- On Fig. 9, assign Ag NPs.

Reviewer #2: This work is having potential data, but no novelty, a simple repeat of already exiting report. The synthesis of AgNPs with plant extract is mushroomed in the literature. The MS, though, having good data, but I could not see any novelty to the field. I would suggest the work shall be modified to focus on Cather biofilm inhibition with standard drugs and other available AgNPs (might be synthesized by different methods).

Further

1. The Fig 10 is inappropriate, require evidence based pathway

2. Light Microscopy and Florescent Microscopy images shall be placed under suppl doc

3. Include CFLSM image for biofilm inhibition

4. TEM is showing a cluster of AgNPs, required scale marked particles

5. Self agglomeration of synthesized AgNPs on storage is required

6. Language and presentation require editing e.g. In the Introduction Pseudomonas is written as Pseudomon as

Reviewer #3: I consider the manuscript is tecnically sound, however some conclusions and discussion must be reconsidered in order to be supported by data obtained, I believe that description of results, the discussion and conclusions are highly restricted by language and strongly suggest a revision by professional editing service.

Detailed minor and major revisions are yellow highlighted in manuscript file attached to revision. in general minor and major revisions are:

Bioengineered phytomolecules-capped silver nanoparticles using Carissa Carandas leaf extract to embed on to urinary catheter to combat UTI pathogens

Minor revisions

All minor revisions are highlighted in manuscript file, these include suggestion for rewrite sentences, and simple changes.

Major revisions

Abstract and introduction

Grammar revision is suggested in some parts of these sections, in manuscript file are highlighted in yellow.

Material and methods

Grammar revision is suggested in some parts of this section, in manuscript file are highlighted in yellow.

Synthesis and optimization of AgNPs production

Include units of Ag ion concentration, volume of leaf extract, etc.

Antibacterial activity

I suggest modification of titles and subtitles order, and include some methodology description described in other method section.

Include description about how the AgNPs concentration was calculated.

Biofilm inhibition assay

Indicate concentration of AgNPs in concentration units (i.e. mg/L) instead of volume units. If cocnetration and volume of AgNps are equivalent please indicate and explain

In Section 2.12 it is not clear the objective of this experiment, please justify.

Results

I suggest to maintain the same subtitles used in methods section in order to establish an order and accordance between methods and results

I suggest include images of AgNPs suspensions obtained at different synthesis conditions (i.e. varying pH, leaf extract concentration, time reaction and Ag ions concentration)

I consider it is necessary to provide clear description of parameters used in each optimization condition of results obtained and presented in fig 1.

I considered necessary to clearly indicate which are the optimal parameters selected for AgNPs synthesis and criteria used for the establishment of these parameters.

It is not clear how the average size of AgNPs observed by HR-TEM was calculate, please include description.

I suggest to include information about how the MICs were calculated?

The fig 4 shows an important inhibition of bacterial growth (O.D.) at 160 mg/L however higher concentration must be proved in order to establish the MICs. I suggest include O.D. measurements of cultures exposed to higher concentrations of AgNPs to obtain a 100% of growth inhibition and establish the MICs

Description of results obtained by SEM must be wide described based on the results presentes in figure 9.

I suggest that the section of results 3.10 (Mechanisms of antibacterial and antibiofilm activity of AgNPs) must be eliminated and included and well describer in discussion section.

Discussion

I suggest general revision of grammar of this sections, some parts of the text are not understandable. (yellow highlighted)

Lines 328-329

Question: With SPR intensity do you refer to intensity in colour? or intensity of the peak absorption in spectra? if you refer to the color, you must provide the images of AgNPs suspensions . if you refer to the absorption peak, in figure 1a a variation of peak intensity and wavelenght of maximum absorption was clearly observed, thus an effect of pH in the intensity of absorption peak is produced.

Lines 327-340

I consider that based on FTIR results, probable phytomolecules involved in stabilization and capping of AgNPs must be provided and make a comparison with results obtained in previous studies on which phytosynthesis of AgNPs was carried out.

Lines 346-347

I consider is important to indicate how the particle size average was determined, HR-TEM indicate certain grade of heterogenicity of particle size, and in this part of discussion you describe that AgNPs are homogeneous in size, however in conclusion section a size heterogeneity of AgNPs was mentioned. Please describe results, discussion and conclusion according to the data obtained.

Line 358

I consider that a wide discusion based on the scientific litterature about the effciency of AgNPs coated catheters against UTIs must be provided.

Line 378

I consider that a wide description of the figure 10 was necessary, adapt the information provided below to the mechanisms described in figure.

Line 386-394

I consider that this part of discussion must include comparison of the previous studies described with the results obtained in this work. And include a wide discussion about phytomolecules involved in AgNPs synthesis.

Conclusions

I suggest rewrite the conclusions, cause I consider that some conclusions show discrepancy with the results and discussion, some of this conclusions are not supported by data presented.

Figures and tables

In general I suggest to improve the figure description, in order to be clear, informative and to support the description of the results. Also improve of resolution is recommended.

6. PLOS authors have the option to publish the peer review history of their article (what does this mean?). If published, this will include your full peer review and any attached files.

Reviewer #1: **Yes: **Mohamed Abd Elkodous

Reviewer #2: No

Reviewer #3: **Yes: **LUZ ELENA VIDALES RODRIGUEZ

---

## [Author Response · Author response to Decision Letter 0]

21 Jul 2021

Plos One Journal Modifications 

1. Revised manuscript has been changed to the style requirements of PLOS ONE

2. Tables has been included in the revised manuscript and removed separate file 

3. We didn’t receive any funding for this work so please change it to “The authors received no specific funding for this work”

4. Minimal data set has been included as a supplementary file.

5. The figure 10,11 is similar but not identical to the original image and is therefore for illustrative purpose only and the figure 5 has been changed in the revised manuscript.

Response to reviewers comments

We are thankful to the Reviewers 1,2, and 3 for their kind and constructive feedback. As suggested by the reviewers, we have changed/addressed the following comments and the same has been highlighted in the revised manuscript with the response to the reviewers’ file. 

No Page/Section Comments by Reviewer #1 Response by the authors

1 Introduction In the introduction, authors should justify why they decided to use Ag NPs and leaves of C. carandas? Highlight their advantages, because we cannot simply use something because just it is available! We have improved the introduction part as per your suggestion. Reviewer can find the improved part at line 76-79 and line 86-94 in the revised manuscript.

2 Line 60 Line 60, "Leaves of C. carandas were used to yield Ag NPs", I think you need to rephrase this sentence, as leaf extract can only be used to stabilize formed Ag NPs and / or reduce the precursor solution of silver nitrate into Ag NPs. We have rephrased the sentence and can be found at line 95-97 of the revised manuscript.

3 Line 93 Line 93, wavelength of Cu-Kα radiation is not correct, the correct value is 1.5406 Å Correct value can be found at line 141 in the revised manuscript 

4 Line 225-line 93 In line 225, authors used Scherrer formula to determine crystalline size, and they mentioned non-correct wavelength in

Line 93, then accordingly, the calculated size will not be correct. Please check this size again. The wavelength has been corrected in line 141 of revised manuscript. Therefore, size mentioned in the line 313 of revised manuscript doesn’t need any modification 

5 XRD pattern contains non-assigned peaks, please explain. Detailed description was made and can be found at line 316-320in the revised manuscript 

6 on FTIR spectra, it is better to highlight, peaks confirming the conjugation between Ag NPs and the extract Highlighted peaks confirm the capping can be found at Fig 4 D in the revised manuscript 

7 On SAED pattern, you should assign the crystalline planes and match them with those obtained by XRD. Fig 4 C of the revised manuscript shows the marked diffraction rings corresponds to the peaks obtained in XRD 

8 Fig.2 Fig. 2 is not clear; it is better to draw the data using suitable software Suggested modifications were done in the revised manuscript and can be found as Fig 2 and Fig 3 

9 Fig. 3 Fig. 3 it is hard to see the label, also indicate the ZOI on the figure for each tested sample. Suggested modification are done in the revised manuscript and can be found as Fig 4 and Fig 5 

10 Fig.4 Fig.4, error bars should be added Suggested modification are done in the revised manuscript and can be found as Fig 7 

11 Fig. 9 On Fig. 9, assign Ag NPs. Suggested modifications are done in the revised manuscript and can be found as Fig 10

No Page/Section Comments by Reviewer #2 Response by the authors

1 Fig 10 The Fig 10 is inappropriate, require evidence-based pathway The actual mechanism was not found through our study but we are coming up with the mechanism already available in the literature and we have changed the text in figure instead of Carisa carandas AgNPs it is mentioned as plant AgNPs and also, we have widely discussed about the biofilm mechanism in the discussion part line 545-564

2 Light Microscopy and Florescent Microscopy images shall be placed under suppl doc It is placed under supplementary file as per your suggestion and can be found as Supplementary document in the revised manuscript

3 Include CFLSM image for biofilm inhibition As stated in the financial disclosure this study does not have any funding it is very hard for us to afford this imaging as it is not available in our institutions. However, we will try to sort out this issue in the future studies. 

4 TEM is showing a cluster of AgNPs, required scale marked particles Suggested modifications by the reviewer has been done and can be found at Fig 4 (A) in the revised manuscript 

5 Self-agglomeration of synthesized AgNPs on storage is required We have found the AgNPs solution was stable for the period of two months under dark. Hence no agglomeration was taken place in the solution and then we lyophilized the AgNPs to obtain AgNPs powder for the purpose of application. Therefore, no chance of self-agglomeration takes place

6 Language and presentation require editing e.g. In the Introduction Pseudomonas is written as Pseudomon as All the necessary modifications were done in the revised manuscript

No Page/Section Comments by Reviewer #3 Response by the authors

1 All minor revisions are highlighted in manuscript file, these include suggestion for rewrite sentences, and simple changes All the minor revisions were changed according to the suggestion of the reviewer in the revised manuscript

2 Abstract and introduction

Grammar revision is suggested in some parts of these sections, in manuscript file are highlighted in yellow. The grammar revisions were changed according to the suggestion of the reviewer in the revised manuscript 

3 Synthesis and optimization of AgNPs production

Include units of Ag ion concentration, volume of leaf extract, etc. Suggested modifications by the reviewer has been done in the revised manuscript and can be found at line 122-134 

4 Antibacterial activity

I suggest modification of titles and subtitles order, and include some methodology description described in other method section.

Include description about how the AgNPs concentration was calculated. 

Suggested modifications by the reviewer has been done and can be found at line 154-162 in the revised manuscript

5 Biofilm inhibition assay

Indicate concentration of AgNPs in concentration units (i.e. mg/L) instead of volume units. If concentration and volume of AgNPs are equivalent please indicate and explain Suggested modifications by the reviewers has been done and can be found at line 224-226 and at line 237-250 in the revised manuscript

6 In Section 2.12 it is not clear the objective of this experiment, please justify. The experiment title has been changed and the objective has been well described at line 252-256 in the revised manuscript 

7 Results

I suggest to maintain the same subtitles used in methods section in order to establish an order and accordance between methods and results As per the reviewer’s suggestion, we have maintained the same subtitles in methods and results which can be found in the revised manuscript 

8 I suggest include images of AgNPs suspensions obtained at different synthesis conditions (i.e. varying pH, leaf extract concentration, time reaction and Ag ions concentration) As per the reviewer suggestion the image for color of AgNPs synthesis has been added in Fig 1 and Fig 2 (A, B, C, D)

---

## [Decision Letter · Decision Letter 1]

9 Aug 2021

PONE-D-21-17789R1

Bioengineered phytomolecules-capped silver nanoparticles using Carissa carandas leaf extract to embed on to urinary catheter to combat UTI pathogens

PLOS ONE

Dear Dr. Saravanan,

Thank you for submitting your manuscript to PLOS ONE. After careful consideration, we feel that it has merit but does not fully meet PLOS ONE’s publication criteria as it currently stands. Therefore, we invite you to submit a revised version of the manuscript that addresses the points raised during the review process.

ACADEMIC EDITOR: Your manuscript can be accepted provided you are ready to undertake minor revision as suggested by reviewer 3.

We look forward to receiving your revised manuscript.

Kind regards,

Amitava Mukherjee, ME, Ph.D.

Academic Editor

PLOS ONE

Journal Requirements:

Reviewers' comments:

Reviewer's Responses to Questions

**Comments to the Author**

1. If the authors have adequately addressed your comments raised in a previous round of review and you feel that this manuscript is now acceptable for publication, you may indicate that here to bypass the “Comments to the Author” section, enter your conflict of interest statement in the “Confidential to Editor” section, and submit your "Accept" recommendation.

Reviewer #1: All comments have been addressed

Reviewer #3: All comments have been addressed

2. Is the manuscript technically sound, and do the data support the conclusions?

Reviewer #1: Yes

Reviewer #3: Yes

3. Has the statistical analysis been performed appropriately and rigorously? 

Reviewer #1: Yes

Reviewer #3: N/A

4. Have the authors made all data underlying the findings in their manuscript fully available?

Reviewer #1: Yes

Reviewer #3: Yes

5. Is the manuscript presented in an intelligible fashion and written in standard English?

Reviewer #1: Yes

Reviewer #3: Yes

6. Review Comments to the Author

Reviewer #1: In the revised version of their manuscript, authors addressed the required comments properly and I think the article is acceptable.

Reviewer #3: Comments has been adressed by authors, however, some mistakes remains in the manuscript, most of those are simple mistakes and easy to correct and has been highlighted in the manuscript file.

Description of an specific part in the discussion section (highlighted in mauscript file) can be improved.

In fig 11, I stronglly suggest to modify the figure, specifically, eliminate some cellular organelles which are specific for eukaryiotic cells.

7. PLOS authors have the option to publish the peer review history of their article (what does this mean?). If published, this will include your full peer review and any attached files.

Reviewer #1: **Yes: **Mohamed Abd Elkodous

Reviewer #3: No

---

## [Author Response · Author response to Decision Letter 1]

13 Aug 2021

PLoS one Requirements

We have included new references to revised manuscript as it is necessary to address the comments by the reviewers and no retracted reference has been added to the revised manuscript and the reference list in the revised manuscript is complete.

Response to Reviewers comments 

We are thankful to the Reviewers 1 and 3 for their kind and constructive feedback. As suggested by the reviewers, we have changed/addressed the following comments and the same has been highlighted in the revised manuscript with the response to the reviewers’ file. 

Reviewer Comments:

 Description of a specific part in the discussion section (highlighted in manuscript file) can be improved.

Response 

All the necessary modifications have been done according to the reviewer comments and can be found in the revised manuscript of line 73-74, line 154-155, line 279-280, line 283, line 294, line 303-305, line 340, line 412, line 450-451, line 498, line 502-503, line 506-514, line 517-522, line 526, line 530-532, line 534-535, line 557-558

Reviewer Comment:

Calculation of the final concentration of AgNPs in this experiment is clear, however, it remains unclear how do you determine the concentration of silver ions of biosynthesized AgNPs suspension (stock suspension of AgNPs)

Response:

We have determined the concentration of silver ion present in the AgNPs suspension based on the Avogadro number (6.023×1023) and the experimental confirmation of the silver ion concentration can be done using atomic absorption spectroscopy. As stated in the financial disclosure this study does not have any funding it is very hard for us to afford this spectroscopy as it is not available in our institutions. However, we will try to sort out this issue in the future studies

Reviewer Comment:

I suggest to discuss these studies in the context of the mechanism proposed

Response

As per the reviewer suggestion we have modified the image and context of the antibiofilm mechanism and can be found in the revised manuscript as fig 12 and at line 568-572, line 579-581

Reviewer Comment:

In fig 11, I strongly suggest to modify the figure, specifically, eliminate some cellular organelles which are specific for eukaryotic cells

Response:

As per the reviewer suggestion we have modified fig 11 and the revised image can be found in the revised manuscript

---

## [Editor Report · Decision Letter 2]

16 Aug 2021

Bioengineered phytomolecules-capped silver nanoparticles using Carissa carandas leaf extract to embed on to urinary catheter to combat UTI pathogens

PONE-D-21-17789R2

Dear Dr. Saravanan,

We’re pleased to inform you that your manuscript has been judged scientifically suitable for publication and will be formally accepted for publication once it meets all outstanding technical requirements.

Kind regards,

Amitava Mukherjee, ME, Ph.D.

Academic Editor

PLOS ONE
---

## [Editor Report · Acceptance letter]

24 Aug 2021

PONE-D-21-17789R2 

Bioengineered phytomolecules-capped silver nanoparticles using *Carissa carandas* leaf extract to embed on to urinary catheter to combat UTI pathogens 

Dear Dr. Muthupandian:

I'm pleased to inform you that your manuscript has been deemed suitable for publication in PLOS ONE. Congratulations! Your manuscript is now with our production department. 

Kind regards, 

on behalf of

Professor Dr. Amitava Mukherjee 

Academic Editor

PLOS ONE